# Novel strategy for wide-range wind vector measurement using the hybrid CP/CTD heating mode and sequential measuring and correcting

Tian Wang[1], Yunbo Shi[1], Xiaoyu Yu[2]*, Guangdong Lan[2], Congning Liu[1]

**1** The Higher Educational Key Laboratory for Measuring & Control Technology and Instrumentations of Heilongjiang Province, School of Measurement-Control Technology & Communications Engineering, Harbin University of Science and Technology, Harbin, Heilongjiang, China, **2** School of Atmospheric Sciences, Sun Yat-Sen University, Zhuhai, Guangdong, China

* sysyuxiaoyu@163.com

**Data Availability Statement:** All relevant data are within the manuscript and its Supporting Information files.

## Abstract

To improve the performance of wind sensors in the high velocity range, this paper proposes a wind measurement strategy for thermal wind velocity sensors that combines the constant power and constant temperature difference driving modes of the heating element. Based on the airflow distribution characteristics from fluid dynamics, sequential measurement and correction is proposed as a method of measuring wind direction. In addition, a wind velocity and direction measurement instrument was developed using the above-mentioned approaches. The test results showed that the proposed instrument can obtain large dynamic wind velocity measurements from 0 to 60 m/s. The wind velocity measurement accuracy was ±0.5 m/s in the common velocity range of 0–20 m/s and ±1 m/s in the high velocity range of 20–60 m/s. The wind direction accuracy was ±3˚ throughout the 360˚ range. The proposed approaches and instrument are not only practical but also capable of meeting the requirements of wide-range and large dynamic wind vector measurement applications.

## Introduction

Wind field measurement, i.e. wind speed and direction measurements, provides essential reference information for studying the operation of the atmospheric circulation, water vapor transport conditions and other natural environments. Wind field measurement is also widely used in some artificial environments such as mines, warehouses, buildings and other ventilation systems [1–6]. The wind field measurement for natural environment is characterized by large wind speed dynamics and fast wind direction changes [7, 8]. While in artificial environment, accurate wind speed and direction measurements are needed even the wind speed is small and the wind direction is stable [9, 10].

The wind distribution in outdoor and semi-outdoor spaces such as open hangars and semi-open equipment rooms or warehouses is strongly influenced by natural wind fields. Such

**Funding:** This study was financially supported by the National Key Research and Development Project (No. 2016YFA0602701) and National Key Scientific Instrument and Equipment Development Project (No. 42027804). The funders had no role in study design, data collection and analysis, decision to publish, or preparation of the manuscript.

spaces are also affected by various artificial factors such as cargo placement and building structure. Hence, the wind distribution in these locations differ from that in the natural environment [11–14]. In such cases, accurate wind velocity measurement should be performed at low and medium velocity ranges, while also covering the high velocity range. Furthermore, wind direction should be determined to analyze wind distribution in detail [15].

The common types of wind velocity sensors include thermal, ultrasonic, pitot tube, and mechanical sensors [16]. Mechanical wind velocity sensors are accurate in the high velocity range but not in the low velocity range owing to their starting velocities [17]. Ultrasonic wind velocity sensors are easily affected by the ambient temperature as well as airflow density changes caused by dust or other gases [18–20]. As pitot tube sensors have large volumes, it is difficult to set them up for wind direction calculation [21]. Further, although thermal wind velocity sensors are more suitable for wind velocity and direction calculation, they are disadvantageous for high wind velocity measurement because the heating element can reach ambient temperature, which means that temperature difference will no longer reflect wind velocity [22, 23]. Therefore, extending the measurement ranges of thermal wind velocity sensors while retaining their advantages of high accuracy and high sensitivity is a topic that warrants detailed investigation.

To improve the performance of thermal wind sensors in the high velocity range, a hybrid measurement strategy that combines the constant power (CP) and constant temperature difference (CTD) modes of the sensors was developed in this study. A heating coil composed of Ni–Cr alloy was used as the heating component of the wind velocity sensors. A microcontroller utilized this hybrid strategy to drive the coil and calculated the wind velocity in different working modes.

Subsequently, sequential measurement and correction was employed for wind direction measurement. Airflow control structures such as an airflow tunnel and a damper board were introduced and fully verified by performing computer simulations to obtain the optimized manufacturing parameters and wind velocity sensing positions. Advanced 3D printing technology was utilized to implement the airflow tunnel, and a high-precision rotating platform was used to rotate the tunnel. In addition, a fast extreme-value-finding algorithm was utilized to calculate wind direction from multiple wind velocity sensors. Finally, based on the methods described above, a wind velocity and direction measurement instrument was developed considering the perspectives of the structure, hardware circuit, and software algorithm.

The contributions and novelty of this study can be summarized as follows:

1. A hybrid strategy was developed to drive the heating component by combining the CP and CTD modes. In this strategy, the two modes are dynamically switched by the microcontroller under different measurement conditions. In the low and medium velocity ranges, the sensors operate in CP mode, whereas in the high velocity range, the sensors operate in CTD mode. Furthermore, the wind velocity calculation method switches according to the heating mode.

2. Sequential measurement and correction was utilized as a wind direction measurement method. In this approach, wind direction is measured and corrected at the positions at which the airflow tunnel is parallel and perpendicular to the external wind vector to improve the accuracy of wind direction measurement.

3. A fast extreme-value-finding algorithm was employed to discriminate the extreme output values of multiple wind velocity sensors. This algorithm is a real-time algorithm that can be used to obtain the relative position between the airflow tunnel and external wind vector rapidly, significantly reducing the measurement time.

## Materials and methods

### Wind velocity measurement

King derived the following equation, which serves as the foundation for the study of thermal anemometers [24]:

$$P = (A + BV^{0.5}) \times dT, \tag{1}$$

where $P$ is the heating power of the heating component, $A$ and $B$ are parameters related to the environment surrounding the system, $V$ is the wind velocity, and $dT$ is the temperature variation of the heating element. When the test environment is fixed, $V$ is related to two factors, namely, $P$ and $dT$. Therefore, there are two methods of wind velocity measurement: making $P$ constant, in which case $V$ is related to $dT$, or making $dT$ constant, in which case $V$ is related to $P$. Accordingly, there are two operating modes of the heating component for a thermal wind velocity sensor: CP mode and CTD mode [25, 26].

It is relatively simple to measure wind velocity when the heating component is operating in CP mode. In this case, the wind velocity can be indirectly obtained by measuring the temperature change of the heating component. Thus, the wind velocity sensor is quite sensitive in the low and medium velocity ranges [27]. However, when the wind velocity is too high, the heating component cannot generate sufficient heat. Hence, its temperature will be approximately equal to the ambient temperature, which makes the temperature difference nearly zero. For large wind velocity measurement, the heating component requires a higher driving power and continuously operates at high temperature [28]. These characteristics not only result in energy wastage, but also contribute to quality degradation and the risk of fire. Therefore, CP mode is not suitable for large wind velocity measurement [29]. In CTD mode, the heating component works slightly above the ambient temperature, and the heating power is dynamically controlled to maintain a constant temperature difference [30, 31]. Therefore, to ensure accuracy at low and medium wind velocities, as well as to expand the wind velocity measurement range, the CP and CTD modes are combined in the proposed method. When measuring low and medium wind velocities, the heating component operates in CP mode, whereas at high wind velocities, it operates in CTD mode.

The expected wind velocity measurement range is 0–60 m/s. Across such a large dynamic range, small wind velocity sensors such as micro-heating bulbs or wires or microelectromechanical system sensors cannot satisfy the heating conditions because they cannot withstand large heating powers or high temperatures [32]. To meet the power and temperature requirements, Ni–Cr coils were used as the heating components of wind velocity sensors, and studies are conducted to implement wind velocity sensors with a wide measurement range and large dynamics.

**Hybrid CP/CTD mode strategy for wind velocity measurement.** When a constant voltage or constant current is applied to a coil, without any feedback from the coil temperature, the coil ideally receives constant heating power and works in CP mode. When the coil is in CTD mode, feedback from the coil temperature to the heating power must be established to maintain a CTD. Regardless of whether the heating coil works in CP or CTD mode, a temperature sensor thermally coupled to the coil is required to measure the actual coil temperature. The corresponding electrical quantity is the voltage signal $U_S$ output from the analog front-end circuit after the temperature sensor. The driving power of the coil can be achieved by changing the driving voltage $U_d$.

The following hybrid strategy was established by combining the CP and CTD modes:

$$U_d = \begin{cases} U_0, & T > T_0 \\ U_0 + U_C(k), & T \leq T_0 \end{cases}, \tag{2}$$

where $U_0$ is the preset heating voltage of the coil in CP mode, $U_C(k)$ is the controller output voltage for heating power compensation, $T$ is the measured temperature of the coil, and $T_0$ is the critical temperature of the coil, representing the switching point between the CP and CTD modes. Eq (2) implies that when the wind velocity is in the low or middle range, $T$ is greater than $T_0$, the controller applies a constant voltage to the coil, and the coil works in CP mode. Conversely, when the wind velocity is in the high range, $T$ is less than or equal to $T_0$, the controller drives the coil with a variable voltage, and the coil works in CTD mode.

Further, when the coil is in CTD mode, its temperature can be controlled using a digital proportional–integral–derivative (PID) algorithm [33, 34]. The power compensation given by the controller can be expressed as

$$U_C(k) = K_p e(k) + K_i \sum_{j=0}^{k} e(j) + K_d[e(k) - e(k-1)], \tag{3}$$

where $U_C(k)$ is the power compensation voltage from the PID controller and $e(k)$ denotes the control deviation, which is the difference between the actual and target compensation voltages. The target voltage is a multiple of the coil temperature output voltage $U_S$, providing negative feedback adjustment of the coil temperature and heating power. Further, $K_p$, $K_i$, and $K_d$ are proportional, integral, and differential coefficients, which are the parameters requiring adjustment in the actual working environment.

**Selection of heating power $P_0$ and critical temperature $T_0$ in CP mode.**   To study the heat loss of the coil at different wind velocities, ANSYS finite element simulation software was used to simulate the thermal situation of the coil to obtain $P_0$ in CP mode and to study the power compensation range of the coil at different wind velocities in CTD mode.

According to the Beaufort wind scale [35], the wind velocity on land is typically grade 0–8, which corresponds to 0–20 m/s. In special cases, the wind velocity will reach grade 18, which corresponds to 60 m/s or more. Thus, the wind velocity sensors should be able to sense wind velocities of 0–20 m/s with high accuracy and 20–60 m/s with acceptable accuracy. Therefore, a wind velocity of 20 m/s can be regarded as the maximum value that can be measured in CP mode. The coil temperature should be higher than the ambient temperature, and it can be regarded as the critical temperature $T_0$ between the two heating modes. This critical temperature could be the boundary condition, and simulations were conducted both above and below 20 m/s. For simulations under the wind velocity of 20m/s, the relationship between the wind velocity and the coil temperature were discovered under the constant heating power; and for above the wind velocity of 20m/s, the relationship between the wind velocity and the ciol heating power under the constant coil temperature were discovered.

Step 1: $T_0$ was found with $P_0$ in CP mode. The heat generation rate of the coil was adjusted, and the simulation was run again at 20 m/s airflow. Nine temperature monitoring points were placed near the center of the coil, and the average of the values obtained at the monitoring points was taken as the temperature output. Once the output was higher than the ambient temperature, it was regarded as the critical temperature $T_0$, and the heat generation rate at this point was considered to be $P_0$ in CP mode, which could be converted into watts. The simulation showed that the heat generation rate was $2.01 \times 10^9$ W/m$^3$ for a coil temperature of 86°C, the heating power was 5.16 W, and the corresponding driving voltage was 11.43 V as the resistance of the coil was 25 Ω. At this heat generation rate, the heat distribution inside the coil was investigated at wind velocities of 0–40 m/s, as shown in Fig 1.

The black circular area in Fig 1 indicates the coil, the contour plot represents the temperature distribution around the coil, and the colored arrows denote the external wind vectors.

 

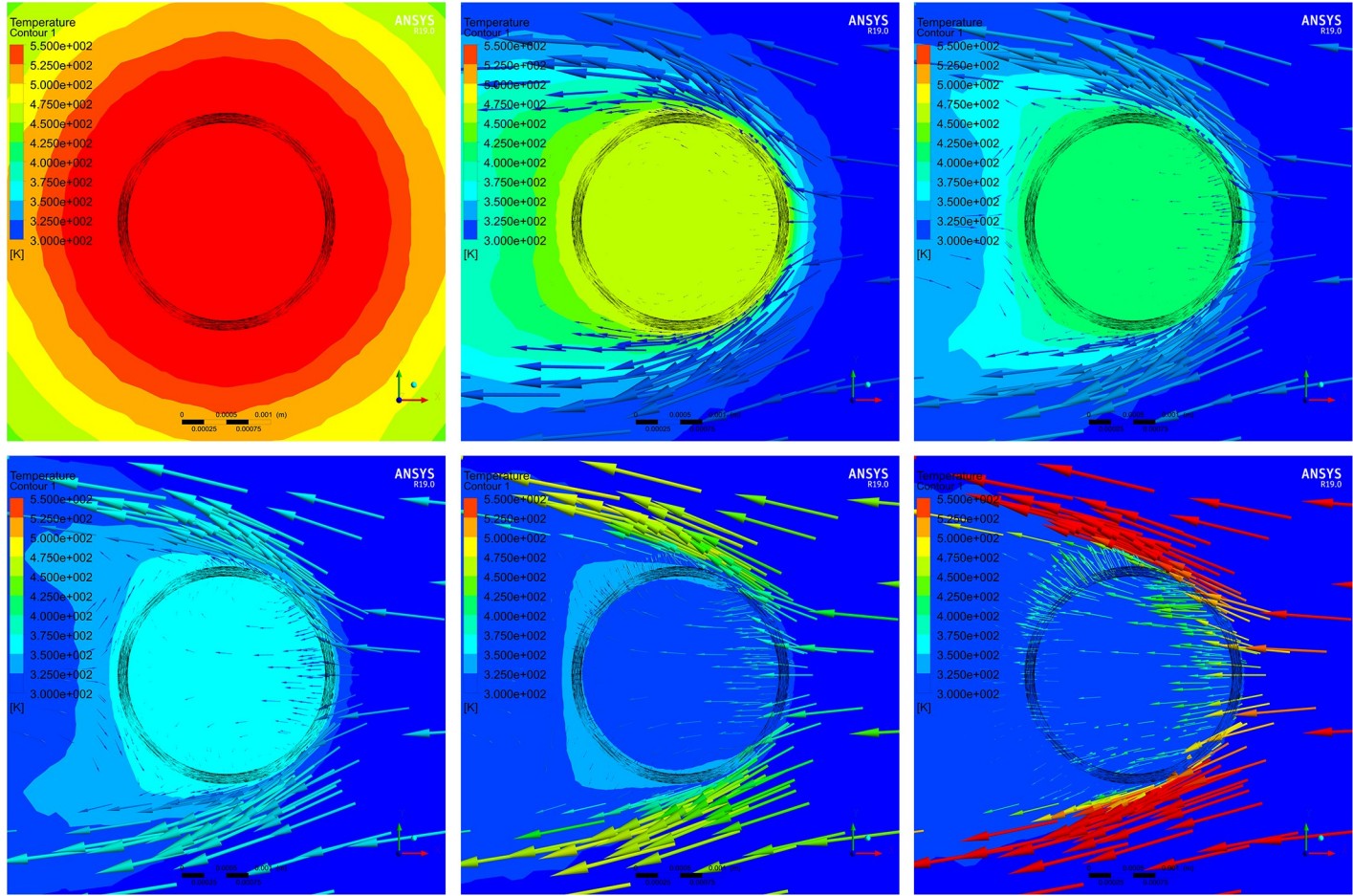

**Fig 1. Heat distribution inside the coil working in CP mode under different wind velocities.** Results for wind velocities of (A) 0 m/s, (B) 5 m/s, (C) 10 m/s, (D) 20 m/s, (E) 30 m/s, and (F) 40 m/s.

Table 1 summarizes the correspondence between the temperature and wind velocity in the center of the coil in CP mode, as obtained from the simulations.

The heat distribution inside the coil is uniform under any wind velocity. In the case of zero wind velocity, the temperature in the center of the coil is approximately 281˚C. When the wind velocity changes, the temperature decreases rapidly and non-linearly. At 20 m/s, the temperature is approximately 86˚C. At 20–40 m/s, the temperature does not change significantly, and the lowest temperature is 45.9˚C, which is close to the ambient temperature and can hardly reflect the airflow velocity.

**Table 1. Wind velocity and temperature of the center of the coil in CP mode.**

| Wind Velocity m/s | Temperature | Wind Velocity | Temperature | Wind Velocity | Temperature |
|---|---|---|---|---|---|
| | ˚C | m/s | ˚C | m/s | ˚C |
| 0 | 281.3 | 15 | 104.5 | 30 | 61.1 |
| 2.5 | 223.4 | 17.5 | 94.4 | 32.5 | 57.1 |
| 5 | 182.7 | 20 | 85.9 | 35 | 52.9 |
| 7.5 | 154.2 | 22.5 | 78.6 | 37.5 | 49.1 |
| 10 | 133.4 | 25 | 72.3 | 40 | 45.9 |
| 12.5 | 117.1 | 27.5 | 66.7 | | |

**Table 2. Wind velocity and coil heating power in CTD mode.**

| Wind Velocity m/s | Power | Wind Velocity | Power | Wind Velocity | Power |
|---|---|---|---|---|---|
| | W | m/s | W | m/s | W |
| 20 | 5.14 | 34 | 7.44 | 48 | 9.64 |
| 22 | 5.47 | 36 | 7.75 | 50 | 9.95 |
| 24 | 5.81 | 38 | 8.11 | 52 | 10.23 |
| 26 | 6.14 | 40 | 8.41 | 54 | 10.49 |
| 28 | 6.45 | 42 | 8.72 | 56 | 10.74 |
| 30 | 6.78 | 44 | 9.00 | 58 | 11.05 |
| 32 | 7.08 | 46 | 9.34 | 60 | 11.30 |

Step 2: The power compensation of the coil was found by maintaining its temperature at 86°C with a wind velocity of 20–60 m/s in CTD mode. By adjusting the power of the coil, i.e., changing the heat generation rate in the simulation software under different wind velocities, the temperature at the center of the coil was maintained at 86°C. The relationship between the wind velocity and coil heating power was obtained by saving the power of the coil at different wind velocities (see Table 2).

In CTD mode, the controller compensates the coil heating power under different wind velocities, which can reflect changes in wind velocity. The higher the wind velocity, the higher the power derived by the coil from the controller. To maintain the coil temperature at 86°C at the maximum wind velocity of 60 m/s, a heat generation rate of $4.42 \times 10^9$ W/m$^3$ is required, which means that the controller should drive the coil with a power of 11.30 W, and the corresponding driving voltage is 16.81 V.

The results of thermal simulations of the coil working under the two heating modes at different wind velocities showed that the hybrid CP/CTD strategy is feasible and has the potential to be implemented. In general, the driving voltage of the coil is low and thus can be provided and controlled with low voltage electronic components. Furthermore, the power of the wind velocity sensor is variable. Usually, the wind velocity is in the low or medium range, and the sensor is driven with low power. When the wind velocity increases, the sensor is temporarily driven with high power. Thus, an optimal balance among the driving power, measurement range, and accuracy can be achieved.

**Heating coil temperature measurement circuit.** A PT100 thermistor was placed into the coil to sense the coil temperature, and the related sensing circuit was designed. The resistance of the PT100 thermistor has the following relationship with the temperature:

$$R_T = R(1 + \alpha T), \tag{4}$$

where $R_T$ is the resistance of the PT100 thermistor at temperature $T$; $\alpha$ is the temperature coefficient, which is 0.00385; $R$ is the standard resistance of the PT100 thermistor at 0°C, which is 100 Ω [36]; and $T$ is the temperature. The resistance of the PT100 thermistor changes linearly with the temperature. If the same current $I$ is applied to the PT100 thermistors for both coil temperature and ambient temperature detection, the resistance change can be converted into a voltage change. By amplifying the difference between the coil temperature and ambient temperature, the effects of the ambient temperature and common-mode voltage were removed [37], as follows:

$$U_O = \beta(U_S - U_A) = \beta I R \alpha (T_S - T_A), \tag{5}$$

where $U_S$ is the voltage output at the coil temperature, $U_A$ is the voltage output at the ambient temperature, and $T_S$ and $T_A$ are the coil temperature and ambient temperature, respectively.

The voltage output $U_O$ is related to the wind velocity. It is advisable to use a small current to drive the PT100 thermistor to avoid self-heating effects. Based on the performance parameters of the PT100 thermistor and the output signal dynamics, $I$ = 1 mA was selected as the driving current of the PT100 thermistor, and the amplification $\beta$ was set to 25 so that the dynamic range of the coil temperature output signal could be expanded to 0–3.3 V, which was suitable for the back-end analog-to-digital converter.

Using the hybrid strategy described above, a detailed circuit and printed circuit board (PCB) were designed and wind velocity sensors were implemented. The wind velocity sensor was a circular PCB with a diameter of 27 mm and height of 10 mm. Fig 2 provides a corresponding circuit block diagram and photograph.

## Wind direction measurement

The heat of the coil in the wind velocity sensors will be carried away by the airflow from all directions. Thus, the coil temperature will decrease, and the wind direction cannot be sensed with only a single wind velocity sensor. Indeed, it is necessary to vary the wind vectors artificially and to calculate the wind direction by measuring and analyzing the wind velocities in different positions. Hence, multiple wind velocity sensors must be placed in suitable positions before calculating the wind direction [38].

An airflow tunnel was designed to vary the wind vectors artificially, control the airflow direction, and use the synchronous measurement of the wind velocity and wind direction. Fig 3 shows the basic method of wind direction measurement using the airflow tunnel.

In the ideal case in which the airflow tunnel is parallel to the external wind vector, the wind will completely pass through the airflow tunnel, and the wind velocity sensors located at the center of the airflow tunnel will be able to sense the wind velocity. When the airflow tunnel is perpendicular to the external wind vector, the wind velocity inside the airflow tunnel will be zero owing to the blockage of the tunnel walls [39]. However, in actual situations, the wind velocity inside will not be zero because of a certain vortex. To reduce the influence of the center vortex, it is necessary to explore the airflow tunnel length under the maximum wind velocity.

In addition, it is only possible to convey the perpendicular or parallel state of the airflow tunnel to the external wind vector using the wind velocity sensor inside the airflow tunnel, whereas it is impossible to convey the direction of the external wind vector. Therefore, it is necessary to use multiple sensors, place the airflow tunnel in an external wind vector field, study the airflow distributions inside and outside the tunnel, and determine the best positions of the wind velocity sensors for obtaining the wind velocity differences. ANSYS Fluent simulation software was used to simulate different cases to investigate the wind velocity distribution

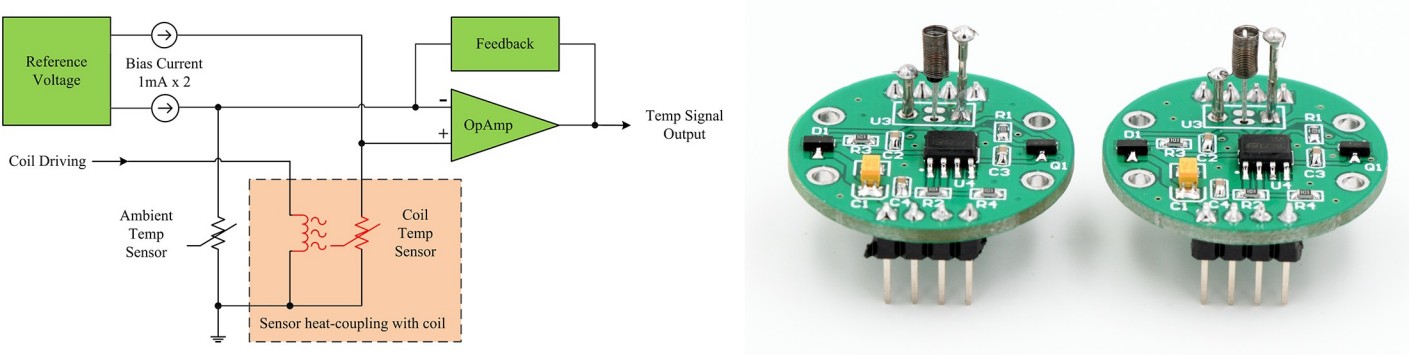

**Fig 2. Wind velocity sensor.** (A) Block diagram and (B) photograph of the sensor.

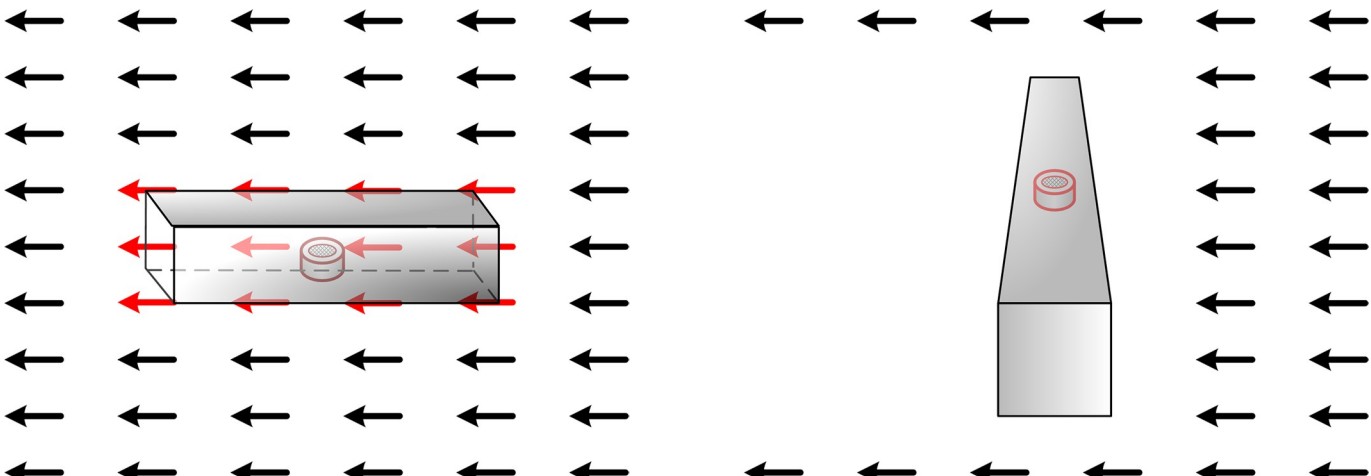

**Fig 3. Basic method of measuring wind direction using the airflow tunnel.** Measurement method when the airflow tunnel is (A) parallel and (B) perpendicular to the wind vector.

of the airflow tunnel when it was parallel and perpendicular to the wind vector and thereby to find the optimal airflow tunnel length and check the sensor positions.

**Airflow tunnel simulation and best positions for wind velocity sensors.** The internal cross-sectional area of the airflow tunnel was selected to be 40×40 mm, which was sufficient to set up the wind velocity sensors, and a certain amount of airflow space was reserved for the wind velocity sensor. Firstly, the optimal airflow tunnel length was studied. The airflow tunnel was placed perpendicular to a wind field with a maximum velocity of 60 m/s, and airflow simulations were performed. The length of the tunnel started at 20 cm and was increased in 1 cm increments. When the central area of the tunnel exhibited a uniform wind velocity distribution that was much smaller than the external wind velocity, the simulation results satisfied with the demand of wind field measurement. Accordingly, the shortest tunnel length with the above-mentioned characteristics is appropriate for achieving adequate performance with the smallest volume and amount of material. Fig 4 presents the simulation results.

The wind velocity distribution in the central area is the smoothest when the airflow tunnel length is 30 cm, and the velocity is much smaller than those of the external wind vectors. When the length increases further, although the low wind velocity area in the airflow tunnel becomes larger, the uneven wind velocity distribution will affect the wind velocity measured by the sensor at the center of the tunnel.

Subsequently, the positions of the airflow control structure and multiple wind velocity sensors were studied to find methods of indirect wind direction measurement. To create a difference in wind velocity or wind direction related to the wind vector angle artificially, a damper board was designed, which was erected at the center of the top of the airflow tunnel and perpendicular to the airflow tunnel [40]. By detecting the difference in wind velocity between two sensors on opposite sides of the damper board, the angle between the airflow tunnel and external wind vector could be calculated. Fig 5 depicts the corresponding structure.

To verify the feasibility of using a tunnel with a damper board for wind direction measurement, the airflow tunnel was 3D-modeled and imported into ANSYS Fluent simulation software. Under the condition of a parallel wind field with a wind velocity of 30 m/s, two airflow distribution views from the y-axis downward were checked. One was the inner center plane of the airflow tunnel, and the other was the damper board plane which was 15 mm above the airflow tunnel.

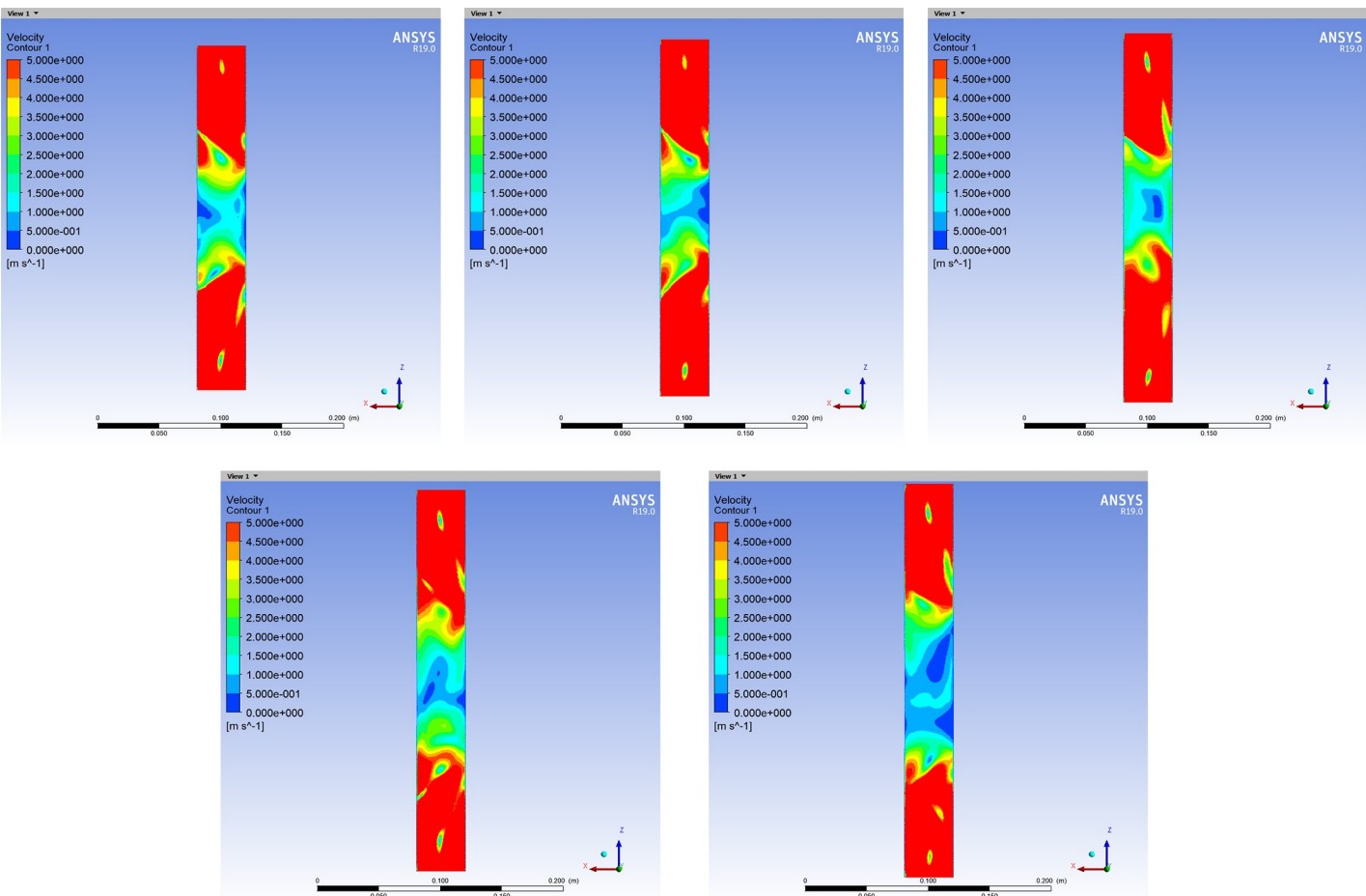

**Fig 4. Wind velocity distribution inside the airflow tunnel when perpendicular to external wind vectors with different lengths.** Results for airflow tunnel lengths of (A) 28 cm, (B) 29 cm, (C) 30 cm, (D) 31 cm, and (E) 32 cm.

Firstly, the airflow distribution in the central plane of the airflow tunnel was investigated. Fig 6 presents the simulation results.

The black frames in Fig 6 indicate the airflow tunnel and damper board, and the green arrows from right to left are wind vectors. The direction and length of each arrow represent the wind direction and wind velocity, respectively. The background is the wind velocity contour graph, where the wind velocity ranges from 0 to 30 m/s and the color gradually changes from blue to red.

According to the simulation results presented above, when the angle between the airflow tunnel and wind vectors is 0˚ or 90˚, the wind distribution inside the airflow tunnel presents a steady state. When the angle is 0˚, the wind vectors completely pass through the airflow tunnel. When the angle is 90˚, the wind vectors are completely blocked by the wall of the airflow tunnel; hence, there is little airflow distribution inside the tunnel, and the wind vector is zero. For other positions between 0˚ and 90˚, the wind distribution inside the airflow tunnel involves obvious turbulence and stratification, which is irregular and cannot accurately reflect the external wind vector information.

Next, airflow distribution on both sides of the damper board was simulated 15 mm above the airflow tunnel. Fig 7 presents the results.

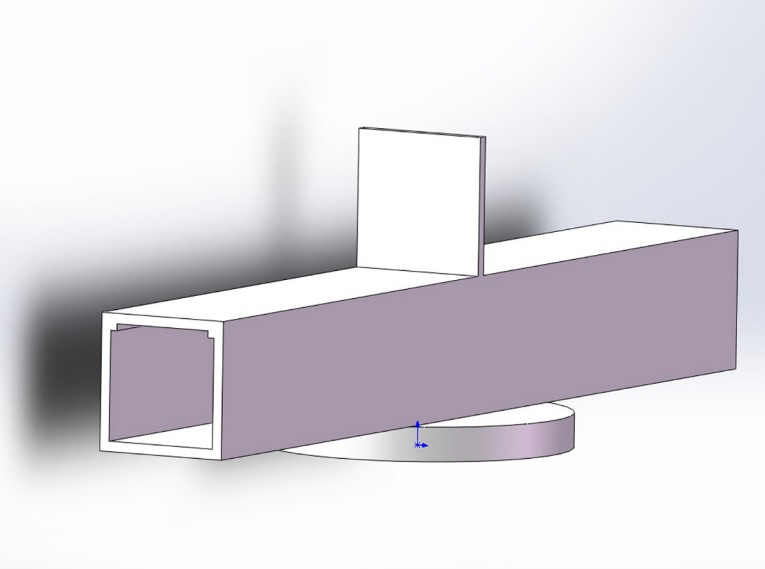

**Fig 5. 3D rendering image of airflow tunnel and damper board.**

It is obvious that the damper board blocks the airflow, resulting in significant variation of the wind distribution on both sides. When the angle between the airflow tunnel and wind vector is 0˚, the wind velocity on the downwind side of the damper board is smaller than that on the upwind side. The wind velocity on both sides of the damper board gradually increases as the tunnel rotates, and the velocity difference between the upper and lower sides decreases. When the angle is 90˚, the damper board has no blocking effect on the wind vector, and the vectors on both sides of the damper board are identical to the external wind vectors. Obviously, the damper board structure will artificially vary the external wind vector, and the wind angle can be measured indirectly by calculating the wind velocity difference between the two sides of the damper board.

According to the simulation results presented above, 300 mm was chosen as the airflow tunnel length. A damper board with a height of 40 mm was installed vertically at the center of the top cover of the airflow tunnel, and two wind velocity sensors were installed on opposite sides of the damper board and close to it to calculate the wind direction. In addition, a wind velocity sensor was installed in the inner center of the airflow tunnel to measure the wind velocity and correct the wind direction. After confirming the structure of the airflow tunnel, ultraviolet (UV)-light curing 3D printing was used to implement all the parts. Fig 8 presents a photograph of the airflow tunnel, damper board, and wind velocity sensor positions.

**Design and implementation of the airflow tunnel, rotating platform, and instrument.** According to the wind direction measurement principles, the airflow tunnel must be rotated smoothly through 360˚, and the rotation angle can be recorded to calculate the wind direction. Hence, the airflow tunnel was installed on a rotating platform with position feedback, and a microcontroller was used as the servo drive.

The microcontroller applied the hybrid CP/CTD strategy to the coils of all the wind velocity sensors simultaneously, drove the coils, read the coil temperatures, and calculated the wind velocities using the coil temperatures and driving powers. The rotating platform was controlled to rotate the airflow tunnel, and the wind direction was calculated according to the wind velocities detected by each sensor. Finally, the calculated wind velocity and wind direction were output to a liquid crystal display screen and uploaded to the host computer through

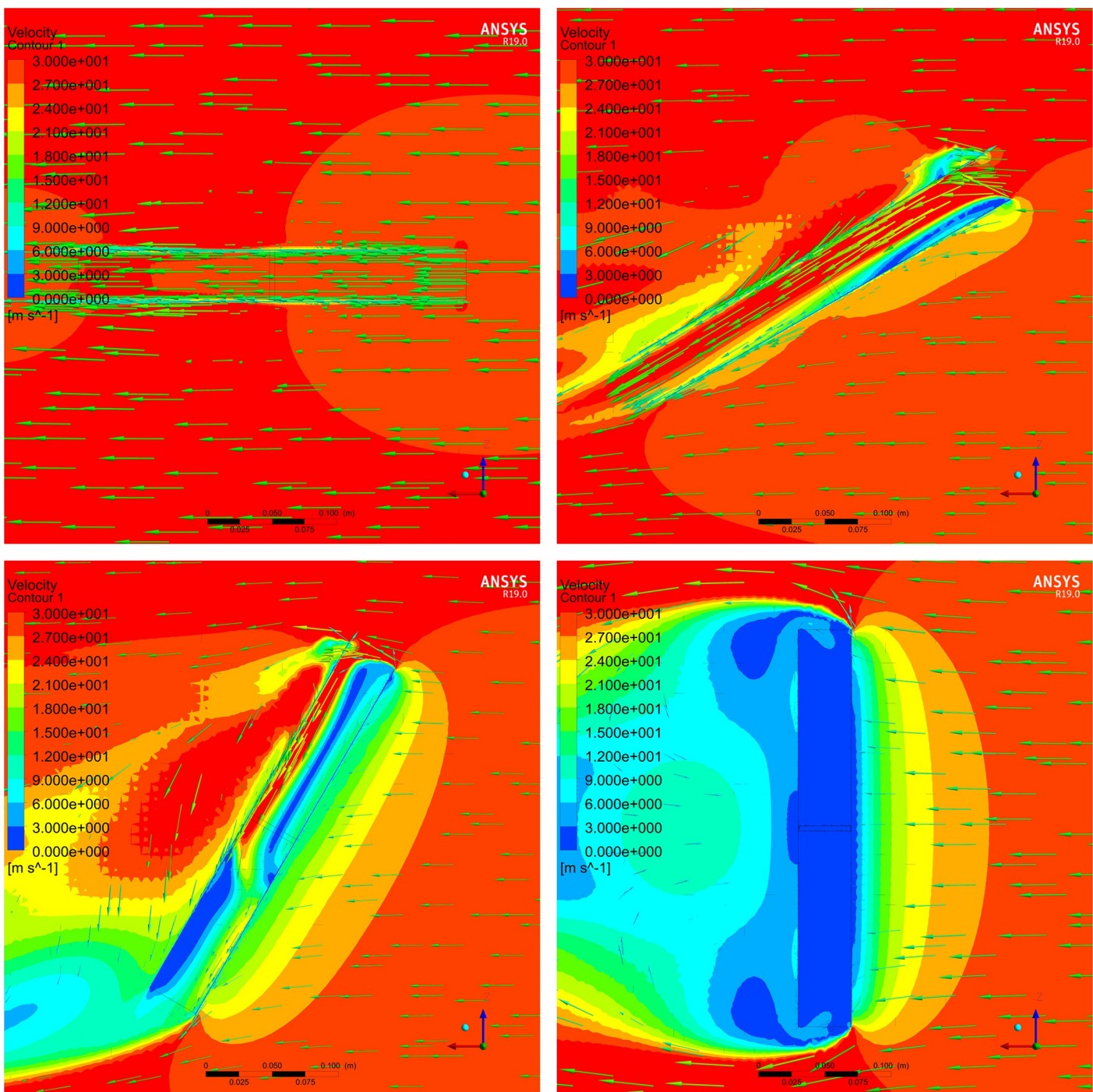

**Fig 6. Airflow distribution of the central plane for different angles between the airflow tunnel and external wind vector.** Results for angles of (A) 0˚, (B) 30˚, (C) 60˚, and (D) 90˚ between the airflow tunnel and external wind vector.

the wireless transceiver for recording. In addition, the host commands were downloaded. Fig 9 shows a hardware block diagram and photographs of the wind velocity and direction measurement instrument.

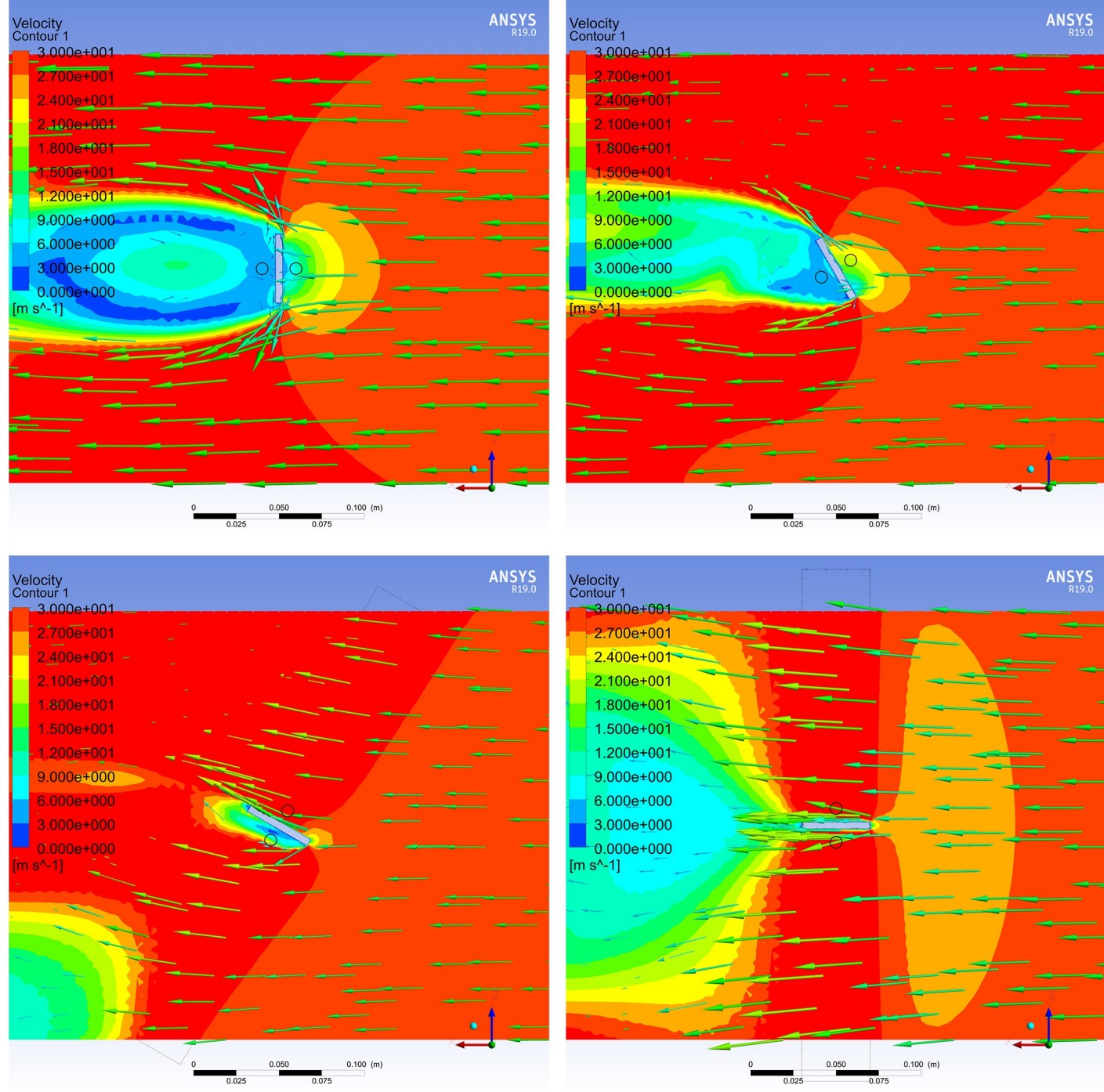

**Fig 7. Airflow distribution of the damper board with different angles between the airflow tunnel and external wind vector.** Results for angles of (A) 0˚ and 90˚, (B) 30˚ and 120˚, (C) 60˚ and 150˚, and (D) 90˚ and 180˚ between the airflow tunnel and wind vector and between the damper board and wind vector, respectively.

## Proposed methods

By combining the basic wind velocity and wind direction measurement principles with the designed mechanical structure and hardware circuit, a complete wind vector measurement procedure and method were developed.

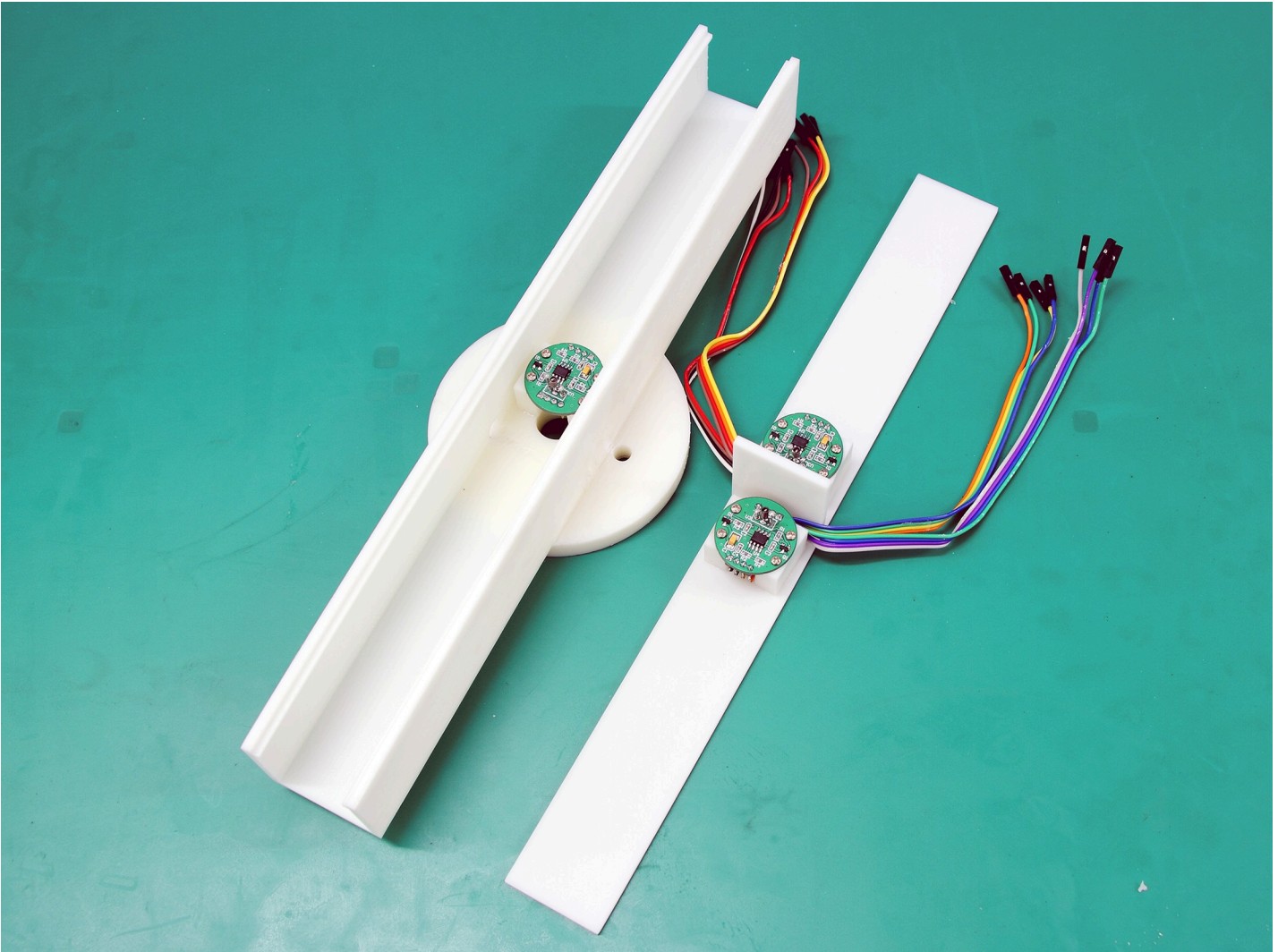

**Fig 8. Image of airflow tunnel, damper board, and wind velocity sensors.**

**General method of wind direction measurement using multiple wind velocity sensors.**
According to the previous simulation results, consider the three airflow tunnel states shown in
Fig 10.

In total, three wind velocity sensors were installed on the surface and inside the airflow tunnel. In state (a), the wind velocity sensors installed on the upwind and downwind sides of the damper board were Sensors #1 and #2, respectively. The sensor installed at the center of the inside tunnel was Sensor #3. In the three states, namely, (a), (b), and (c), of the entire wind direction measurement process, the output values of the three wind velocity sensors changed, as summarized in Table 3.

To measure wind velocity and direction, the angle between the airflow tunnel and wind direction was assumed to be $\theta$, and the outputs of Sensors #1 and #2 were $V_{S1}$ and $V_{S2}$, respectively. There is a certain relationship between the difference in the outputs of the two sensors, $V_{S1}-V_{S2}$, and the change in the difference, together with $\theta$. Specifically, when $V_{S1}-V_{S2}$ is the maximum positive value, $\theta$ is 0˚; when $V_{S1}-V_{S2}$ is the minimum negative value, $\theta$ is 180˚; and

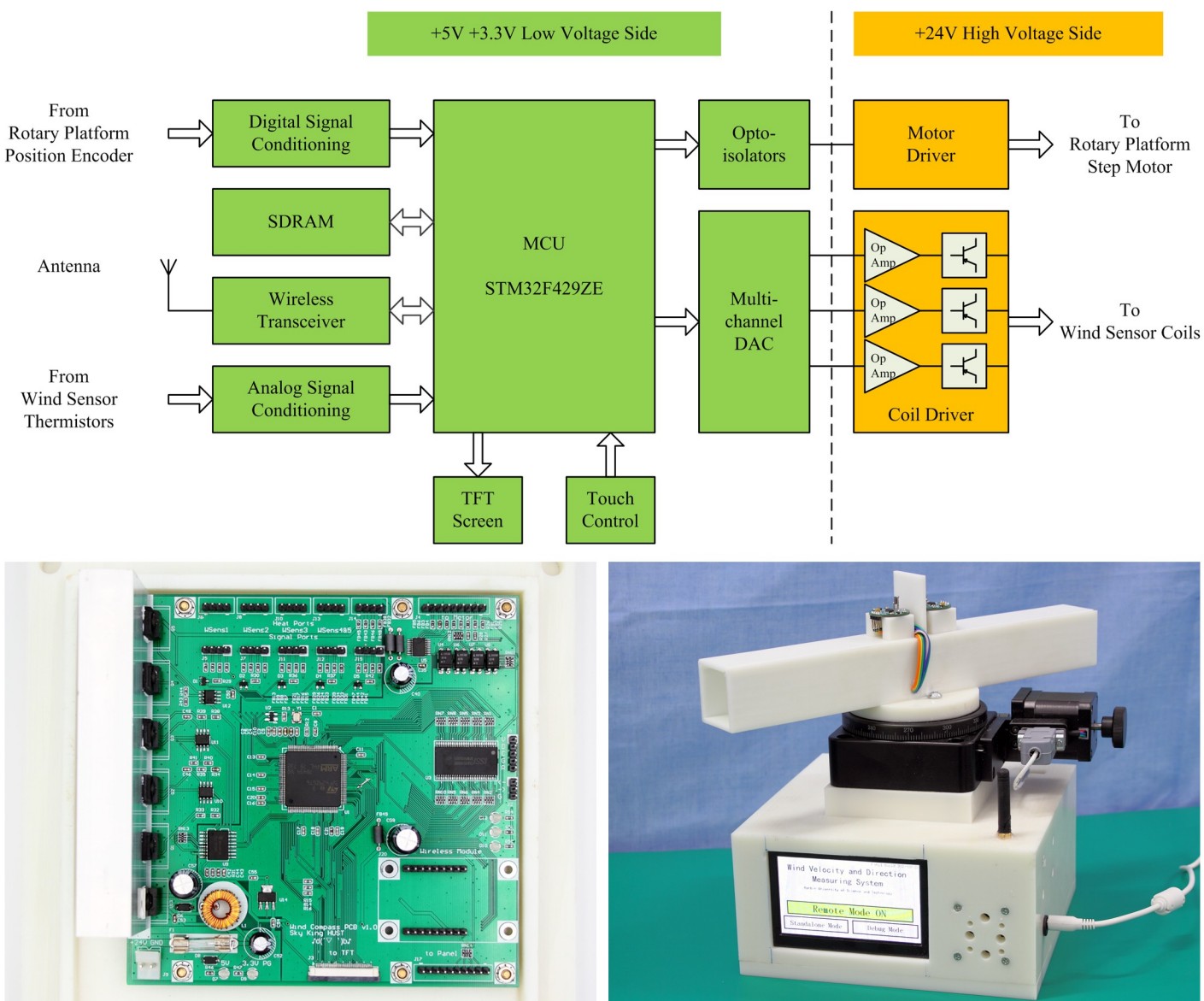

**Fig 9. Wind velocity and direction measurement instrument.** (A) System block diagram (B) PCB (C) Instrument.

when $V_{S1} - V_{S2} = 0$, $\theta$ is 90˚ or 270˚. Sensor #3 outputs the maximum value when $\theta$ is 0˚ or 180˚, and this value can be used as the wind velocity; when $\theta$ is 90˚ or 270˚, it outputs the minimum value, which can be used as an auxiliary correction of wind direction.

**Fast extreme-value-finding method for sensor data based on second-order difference.** According to the requirements of the wind direction measurement method above, the microcontroller must estimate the airflow tunnel position by judging the wind velocities or difference between the velocities. To find the extreme value rapidly, a fast extreme-value-finding method for sensor data based on the second-order difference is proposed.

Assume that $V_S$ is the wind velocity output by any wind velocity sensor, and $\theta$, as defined above, is the angle between the airflow tunnel and wind direction. Then, $V_S$ and $\theta$ have a

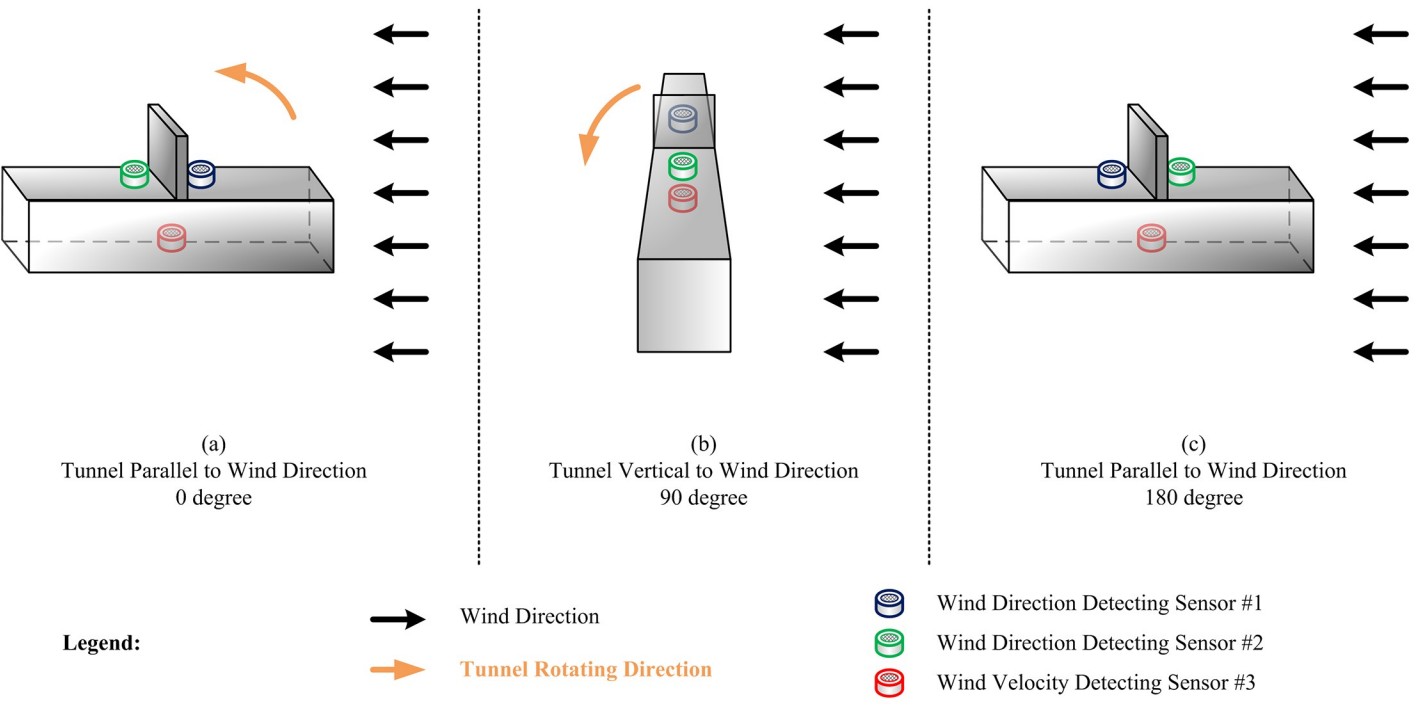

**Fig 10. Complete wind velocity and wind direction measurement process.**

certain relation:

$$V_S = V(\theta). \tag{6}$$

Thus, the first and second derivatives of $V_S$, $V'_S$, and $V''_S$, respectively, can be found. When $V'_S = 0$ and $V''_S > 0$, the minimum value is obtained; when $V'_S = 0$ and $V''_S < 0$, the maximum value is acquired. For microcontroller-based digital systems, continuous calculations must be converted into the discrete field, as follows:

$$D_S[\theta_n] = V_S[\theta_n] - V_S[\theta_{n-1}] \tag{7}$$

$$T_S[\theta_n] = D_S[\theta_n] - D_S[\theta_{n-1}], \tag{8}$$

where $V_S[\theta_n]$ is the $n^{\text{th}}$ sample in the sequential samples of the wind velocity output following the rotation angle $\theta$, and $D_S[\theta_n]$ is the wind velocity increment, i.e., the difference between two adjacent samples in the sequential samples $V_S[\theta_n]$. Further, $T_S[\theta_n]$ performs a differential

**Table 3. Analysis of sensor output in different states.**

| Status | A | Transient A to B | B | Transient B to C | C |
|---|---|---|---|---|---|
| Rotary Angle | 0˚ | 0˚–90˚ | 90˚ | 90˚–180˚ | 180˚ |
| Sensor #1 | Min | Ascending | Max | Descending | Min |
| Sensor #2 | Min | Ascending | Max | Descending | Min |
| #1–#2 | Max, >0 | Descending | = 0 | Descending | Min, <0 |
| Sensor #3 | Max | Descending | Min | Ascending | Max |

calculation between two differentials $D_S[\theta_n]$ and $D_S[\theta_{n-1}]$ again to realize the function corresponding to the second derivative.

However, for digital systems, owing to sampling errors such as quantization error or jitter, it may not be possible to have two samples with zero difference even if the derivative of the continuous function is zero. Therefore, it is acceptable to pre-define a minimum value $E$. When the difference is less than $E$, it can be judged to be zero. Hence, in wind velocity sample series from the sensor following the rotation angle $\theta$ of the airflow tunnel, when there are an angle and samples satisfying

$$D_S[\theta_n] < E \qquad (9)$$

$$T_S[\theta_n] > 0, \qquad (10)$$

the sensor gives the maximum value. When there are an angle and samples satisfying

$$D_S[\theta_n] < E \qquad (11)$$

$$T_S[\theta_n] < 0, \qquad (12)$$

the sensor gives the minimum value.

This extreme-value-finding method can be conducted in real time. The first- and second-order differential values can be calculated during the airflow tunnel rotation. After the extreme value is found, it is unnecessary to continue rotating the airflow tunnel and save redundant data. Therefore, this extreme-value-finding method will significantly improve the measurement speed and reduce the instrument response time during measurement.

**Wind velocity and direction measurement by sequential measurement and correction.** By combining the previous simulation results, wind direction measurement method, extreme-value-finding method, etc., this section describes the complete wind velocity and direction measurement procedure employed in this study using sequential measurement and correction, which was programmed on the microcontroller.

Firstly, after the system was powered on or reset, the microcontroller drove the coils of the wind velocity sensors with the hybrid CP/CTD strategy and waited until all the coils had been fully heated to the working temperature. Next, the microcontroller read the current position of the rotating platform to determine whether it was at the zero position. If the rotating platform was not at the zero position, then it was rotated to the zero position.

Next, the wind velocity and direction measurement were started. The first step was to find the position of the airflow tunnel parallel to the wind vector. The microcontroller drove the rotating platform to make the airflow tunnel rotate counterclockwise from the zero position and recorded the airflow tunnel rotation angle $\theta$ and wind velocity output of Sensor #3, $V_{S3}$. Simultaneously, it read $V_{S1}$ and $V_{S2}$ and found the maximum value of the difference $V_{S1}-V_{S2}$. When the maximum value appeared, $\theta$ was saved as $\theta_H$, representing the position where the airflow tunnel was parallel to the external wind vector. Furthermore, at this position, $V_{S3}$ was saved as $V_H$.

Then, wind direction correction was performed. The microcontroller continued rotating the airflow tunnel counterclockwise and read $\theta$ as well as $V_{S3}$. When the minimum value of $V_{S3}$ was found, $\theta$ was saved as $\theta_V$, representing the position at which the airflow tunnel was perpendicular to the wind vector.

According to the obtained values of $\theta_H$ and $\theta_V$, the rotation angle when the airflow tunnel was parallel to the wind vector was corrected. The difference between $\theta_H$ and $\theta_V$ should be 90°; however, owing to measurement error, the difference was not exactly 90°. Therefore, the

corrected $\theta'_H$ was calculated from $\theta_H$ by taking the mean value of the angle error, as follows:

$$\theta'_H = \theta_H - \frac{90^\circ - (\theta_P - \theta_H)}{2}. \tag{13}$$

After angle correction had been completed, $\theta'_H$ was output together with the previously measured wind velocity $V_H$, completing the wind velocity and direction measurement process. Subsequently, the microcontroller turned the airflow tunnel back to the zero position of the rotating platform and performed the next wind velocity and direction measurement.

## Results and discussion

After the wind velocity sensors and the wind velocity and direction measurement instrument had been implemented, calibrations and experiments were conducted to confirm the system functions and accuracy. The experiments and tests were performed in two parts: wind velocity measurement and wind direction measurement.

### Calibration and testing of wind velocity sensors

The expected wind velocity measurement range was 0–60 m/s. To obtain the airflow to be measured with full-scale adjustable velocity, a high-velocity airflow system was designed and implemented first, as shown in Fig 11.

The airflow system mainly consisted of two axial fans, two cowls, a high-velocity airflow tunnel, a fan controller, and an anemometer. By adjusting the fan controller knob, the rotation speeds of the two fans were adjusted; hence, the air intake and exhaust volumes were adjusted. After the air flowed through the narrow airflow tunnel, high-velocity airflow was generated.

Before each calibration and test of the wind velocity sensors was performed, the wind velocity in the high-velocity airflow tunnel was adjusted using the controller knob and measured using the anemometer. The wind velocity was checked by reading the anemometer and making its value equal the required wind velocity for the current experiment. After checking the wind velocity, the anemometer was removed and a rubber seal was used to close the anemometer hole to ensure the airtightness of the high-velocity airflow tunnel. The wind velocity in the high-velocity airflow tunnel was a predetermined value and was used to calibrate and test the wind velocity sensors.

**Calibration of wind velocity sensors.** The wind velocity sensors worked in the CP and CTD modes for different measurement ranges; hence, the two states were calibrated separately.

When the wind velocity sensors measured wind velocities of 0–20 m/s, they operated in CP mode. According to the simulation results presented above, a constant power of 5.5 W with a voltage of approximately 11.73 V was applied to the coil of the wind velocity sensor from a stabilized power supply unit. The high-velocity airflow generator generated different wind velocities between 0 and 20 m/s in increments of 2 m/s, and the temperature output of the sensor at each wind velocity was saved.

When the wind velocity sensors measured wind velocities of 20–60 m/s, they operated in CTD mode. The microcontroller ran the PID control algorithm and drove the sensor coil, and its output power increased from 5.5 W. As the wind velocity increased, the microcontroller adjusted the power according to the temperature feedback of the sensor. The coil was kept at this operating temperature point, and the wind velocity was calculated from the output power. Therefore, it was necessary to adjust the PID controller parameters first. Fig 12 depicts the PID parameter tuning system.

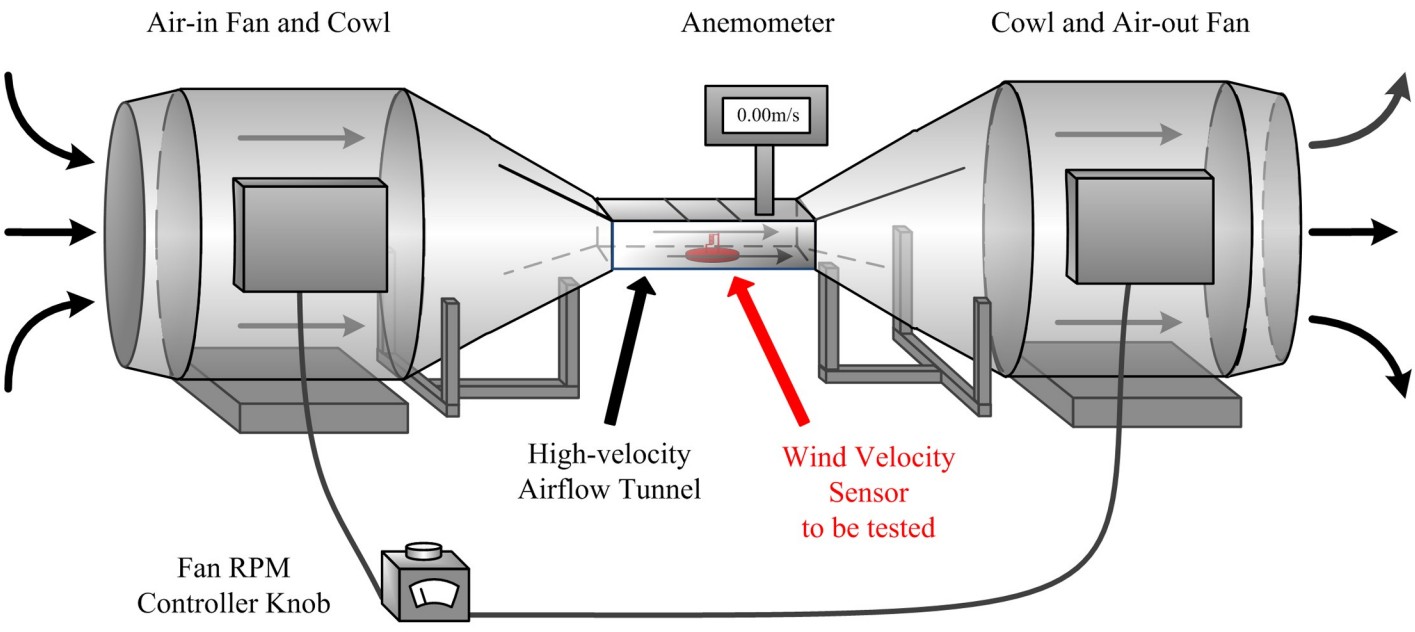

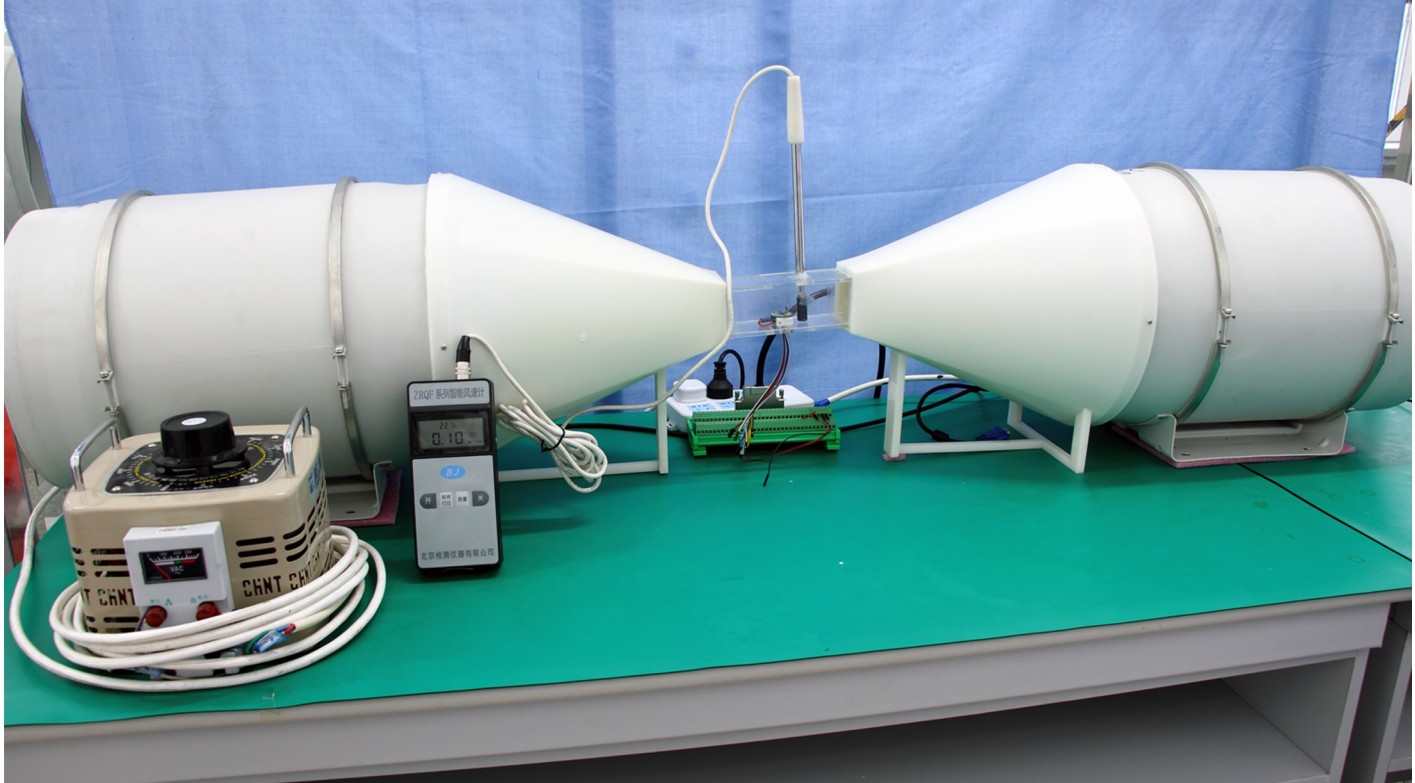

**Fig 11. High-velocity airflow system for calibration and testing of wind velocity sensors.** (A) Block diagram and (B) image of the high-velocity airflow system.

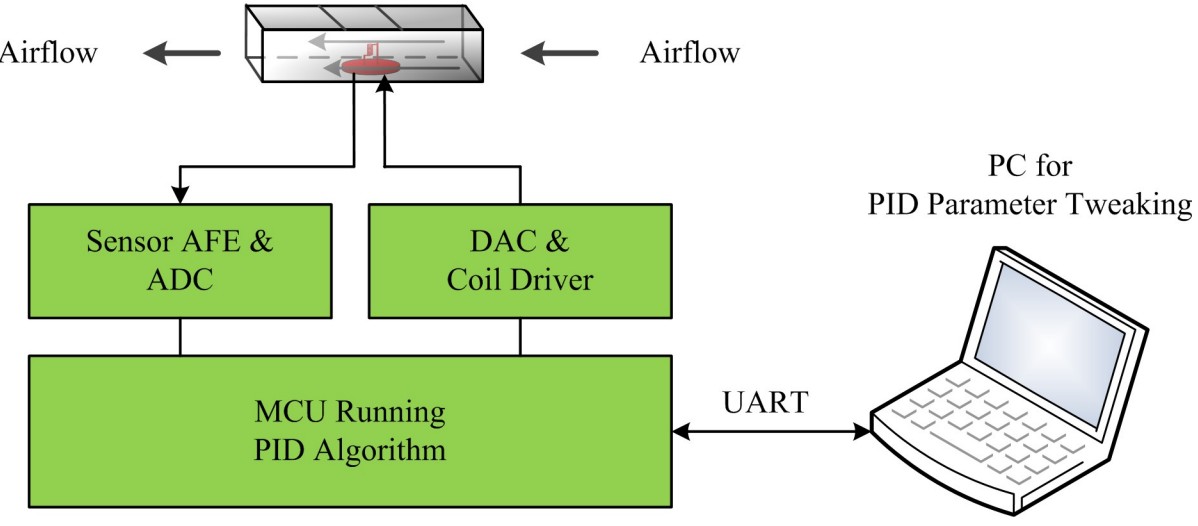

**Fig 12. Schematic diagram of PID parameter tuning system.**

The sensor to be tested was placed in the center of the high-speed airflow tunnel and connected to the PCB of the instrument, so that the microcontroller could read the sensor temperature signal and drive the coil. To facilitate parameter tuning, the host computer ran the PID tuning system using the LabVIEW environment [41, 42]. The microcontroller sent the coil temperature to the host computer and drove the coil using the voltage calculated by the PID tuning system on the host computer through a full-duplex serial interface. The parameters were tuned under a maximum wind velocity of 60 m/s to achieve the largest velocity dynamics so that the measurement system could function well throughout the velocity range. After tuning, the appropriate PID controller parameters were stored in the microcontroller.

Under the PID algorithm control, the coil in the wind velocity sensor worked in CTD mode. To test the sensor, wind velocities ranging from 20 to 60 m/s in 2 m/s increments were generated using the high-velocity airflow generator. At each wind velocity, the heating voltage to the coil was saved.

The temperature and driving voltages of the coil of the wind velocity sensor were measured in the two working modes with different wind velocities, as shown in Fig 13.

When the wind velocity is approximately 20 m/s, the coil temperature voltage is 0.27 V according to the testing data above. Therefore, 0.27 V can be treated as the heating mode switching point of the sensor coil, i.e., the sensor coil is in CP mode when the voltage is greater than 0.27 V, and it is in CTD mode when the voltage is less than or equal to 0.27 V.

To make the discrete measurement points continuous and control the measurement error range, it was necessary to fit the series of measurement points to a polynomial. Furthermore, the measured wind velocity range could be divided into several segments, and a segment function was fitted according to the sensor output voltage so that the relation between the measured and fitted wind velocities could be found for each segment. By writing the polynomial formula for the microcontroller, the measured wind velocity could be corrected in real time with the segmented fitting parameters in the microcontroller, enhancing the accuracy of the wind velocity.

When the sensor is in CP mode, the coil temperature is related to half of the power of the wind velocity, i.e., the wind velocity is related to the square of the coil temperature. Therefore, second-order polynomial regression analysis was performed between the measured wind

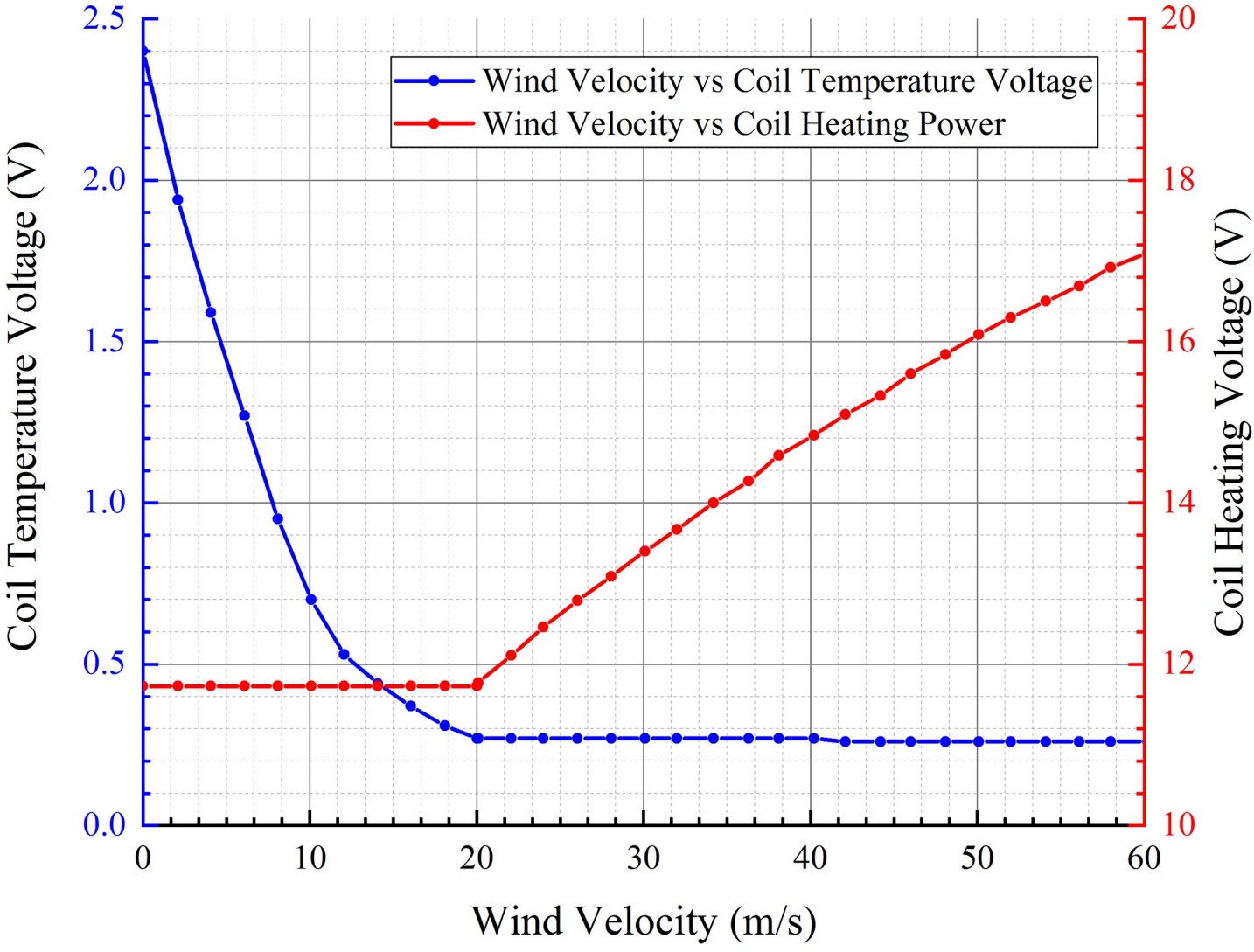

**Fig 13. Temperatures and driving voltages of the coil under different wind velocities.**

velocity and the coil temperature voltage of the sensor. The following regression equation was obtained:

$$\begin{cases} V = 1.023U^2 - 9.088U + 15.91 & \text{When} \quad U \geq 0.7 \\ V = 53.45U^2 - 72.91U + 35.72 & \text{When} \quad 0.27 < U < 0.7 \end{cases}. \tag{14}$$

The wind velocity is also related to the square of the coil heating power when the sensor is in CTD. Hence, second-order polynomial regression analysis was performed between the measured wind velocity and coil driving voltage, and the following regression equation was obtained:

$$V = 0.1138P^2 + 4.461P - 7.957 \quad \text{When} \quad U \leq 0.27. \tag{15}$$

The regression equation parameters were respectively saved in the microcontroller. The microcontroller read the voltage outputs of the wind velocity sensors, judged the sensor

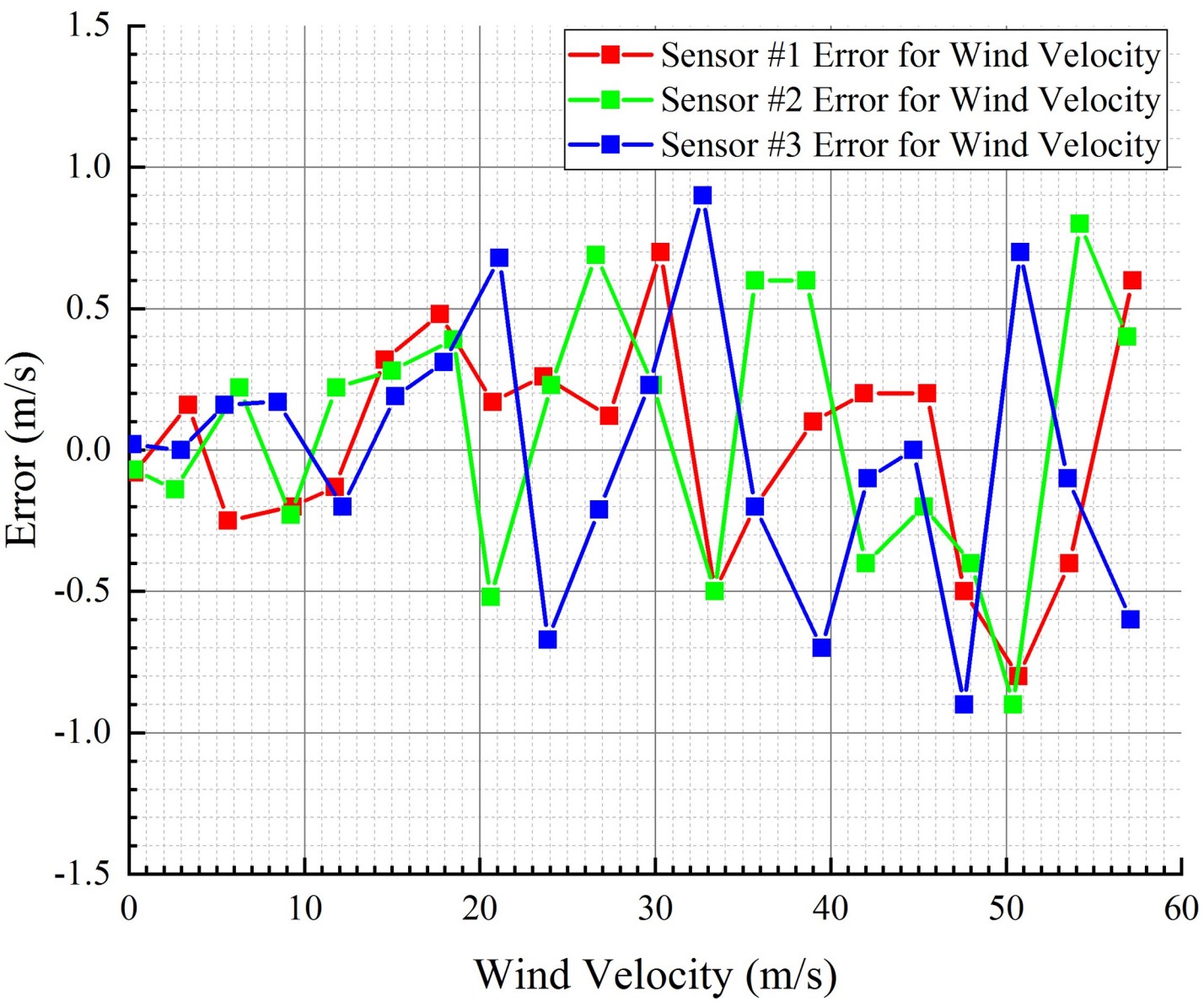

**Fig 14. Error distributions of three sensors.**

working status, and calculated and corrected the wind velocity value according to the coil temperature voltage or coil driving voltage in real time.

**Accuracy and consistency of wind velocity sensors.** To verify the accuracies of the wind velocity sensors after polynomial fitting, as well as the consistency between the different wind velocity sensors, the three wind velocity sensors used in the instrument were tested separately. Each sensor was placed at the center of the high-velocity airflow tunnel of the airflow generator, and 20 wind velocity points within the wind velocity range of 0–60 m/s were randomly selected as the testing wind velocities. The wind velocity from the wind velocity sensor after fitting was compared with that measured by the anemometer to check the measurement error. Fig 14 shows the error distribution of each of the sensors.

In the wind velocity measurement range of 0–60 m/s, an additional 20 wind velocity points were selected to test the consistency of the three sensors. When testing the three sensors with

the same wind velocity point, the RPM control knob of the high-velocity airflow generator was kept still to ensure that the wind velocity was constant, and the three sensors were tested in turn under the same wind velocity. The error between the sensor outputs and standard wind velocity was saved, and the maximum error was used to check the consistency of the three sensors. Fig 15 presents the maximum error among the three sensors.

It can be seen from the error distribution in Fig 15 that the errors between the wind velocity outputs of the three sensors after correction and the standard value meet the design requirements, i.e., within the measurement range of 0–20 m/s, the error is ±0.5 m/s, and within the measurement range of 20–60 m/s, the error is ±1 m/s. It can also be seen from Fig 15 that the measurements of the three sensors are consistent, and the maximum error between the sensors is less than 0.8 m/s. Thus far, the three sensors meet the design requirements; hence, they can be used for wind velocity detection and wind direction calculation.

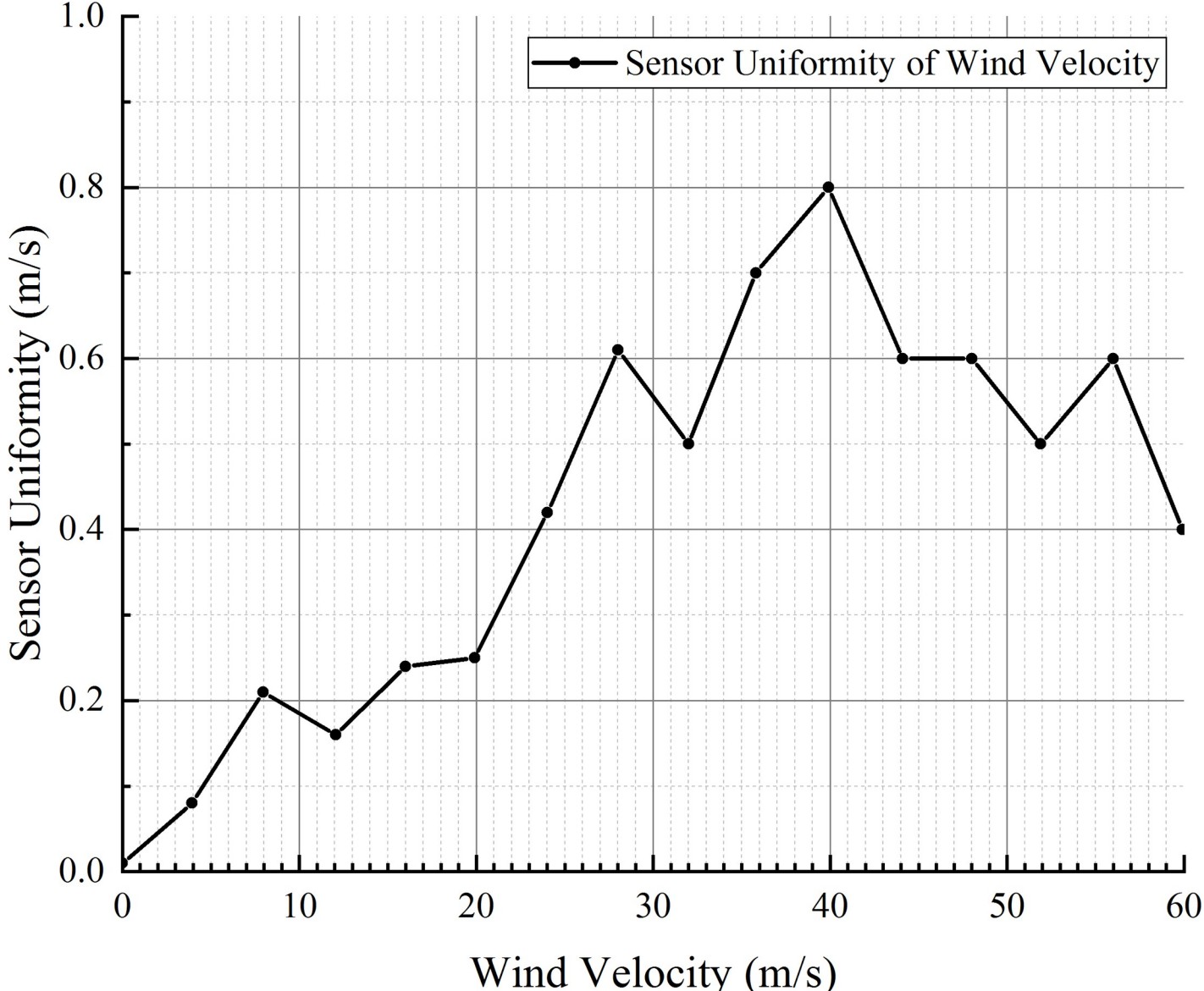

**Fig 15. Consistency of the three sensors.**

## Wind direction measurement test

Next, all parts of the wind velocity and direction measurement instrument were assembled, and the wind velocity sensor calibration data and wind direction calculation method were programmed into the firmware. A command line host client software was programmed using C programming language and run on a host PC so that the wind velocity and direction instrument could be controlled and the wind velocity and direction data could be received and saved.

The relevant tests for wind direction measurement were completed in the wind direction test chamber, as shown in Fig 16.

The wind direction test chamber provided directional airflow through an adjustable-speed fan installed on the side. An angle dial was attached to the floor of the chamber, with a circumference of 360 subdivisions. The wind velocity and direction measurement instrument was placed above the angle dial. Angles in 30˚ increments were taken as the calibration angles in the wind direction test. During each test, the instrument was manually rotated along its center to the calibration angle value, indirectly causing the wind direction to change. To verify the validity of sequential measurement and correction for wind direction measurement, the angles calculated by sequential measurement and correction were compared with those simply calculated from the Sensor #1 and #2 velocities. The wind angle was measured twice: firstly, only Sensors #1 and #2 were used to measure the wind angle; secondly, all three sensors were used to measure the wind angle using the method of sequential measurement and correction. The measured value pairs corresponding to each wind angle were saved and compared to check for errors. Fig 17 presents the angle errors resulting from the two wind angle measurement methods.

According to Fig 17, the maximum angle error range is ±5˚ when performing calculations using only the sensors on opposite sides of the damper board. By adding the third sensor in the center of the airflow tunnel and using the method of sequential measurement and correction, the error of the wind angle decreases significantly from ±5˚ to ±2˚.

Then, 10 additional angles different from the previous calibration angles were selected to test the wind angle measurement and investigate whether the accuracies obtained using those wind angles met the design requirements. Table 4 summarizes the results.

The angle error is less than or equal to ±3˚ in each case, which means that these wind angles all meet the design requirements.

## Conclusion

To overcome the challenges in wind vector measurement over a wide range, a large, dynamic, and high-precision method was developed from the perspectives of the wind sensor coil driving strategy, wind velocity measurement circuit, sensor calibration, and test equipment for outdoor and semi-outdoor spaces such as open tarmacs, semi-open equipment rooms, and warehouses. A prototype of the wind velocity sensor was fabricated and tested. The concept of wind direction measurement by analyzing and fusing a large amount of wind velocity sensor information was considered from the perspectives of the airflow control structure and wind angle calculation algorithm. In addition, related thermal and fluent simulations were performed to find the key parameters and verify the feasibility of the wind direction measurement method. Advanced manufacturing technologies including computer-aided design software and UV-curable 3D printing were introduced, for which the method of sequential measurement and correction as well as a fast extreme-value-finding algorithm were proposed for wind direction measurement. The test results showed that the wind velocity sensors could measure wind velocities of 0–60 m/s, with accuracies of ±0.5 m/s for the 0–20 m/s range and ±1 m/s for

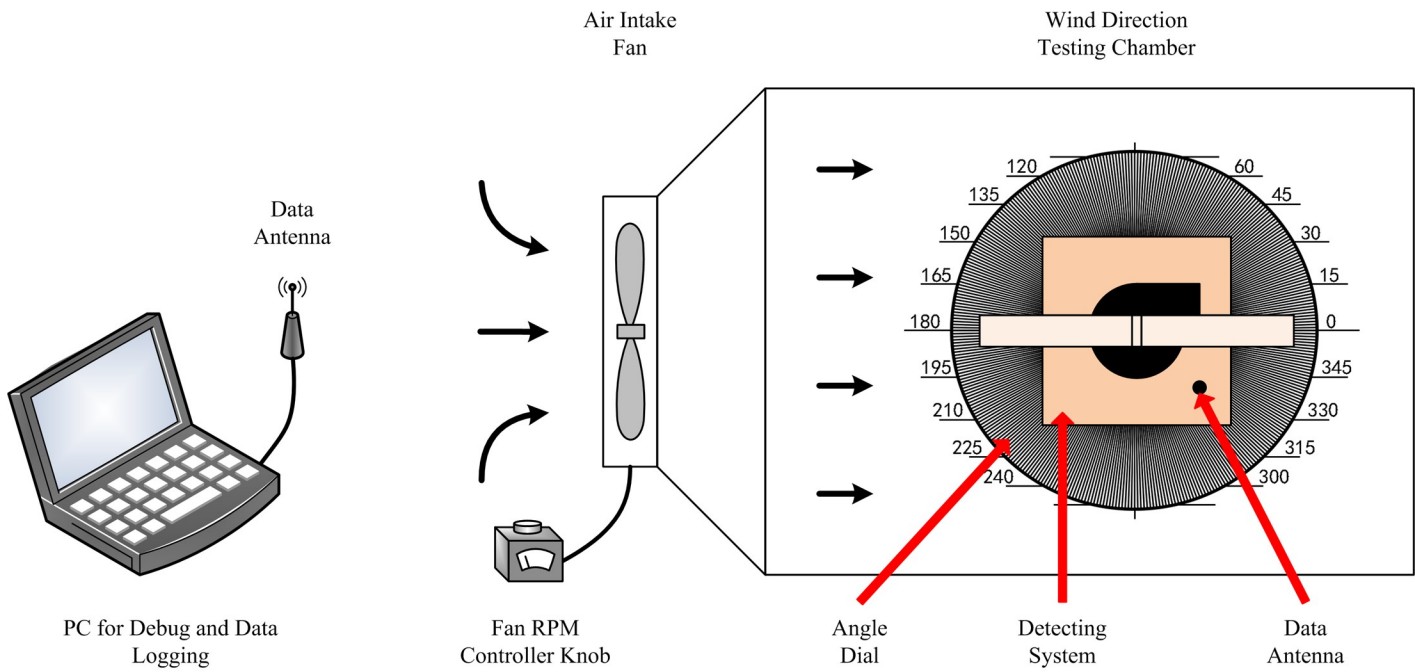

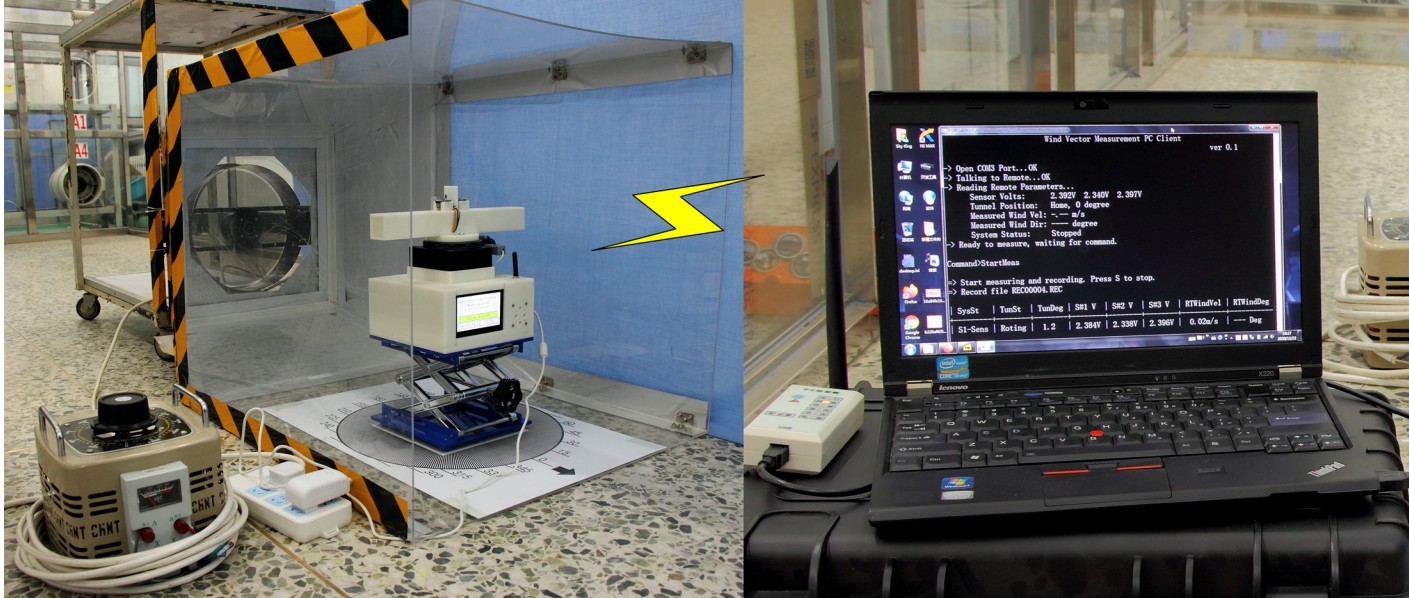

**Fig 16. Wind direction test chamber.** (A) Block diagram and (B) image of the test chamber.

the 20–60 m/s range. For wind direction measurement, the accuracy reached ±3˚ in the full range of 360˚. The wind vector instrument achieved the expected wind velocity and direction measurement accuracies and can be used for applications of wind vector measurement across a wide range with large dynamics and high precision.

There remains scope for improvement of the wind velocity and wind direction measurement accuracies. In this study, the wind velocity information given by multiple wind velocity sensors was only used to calculate the wind direction. By processing the information in detail and applying more advanced methods, the wind velocity measurement accuracy can be

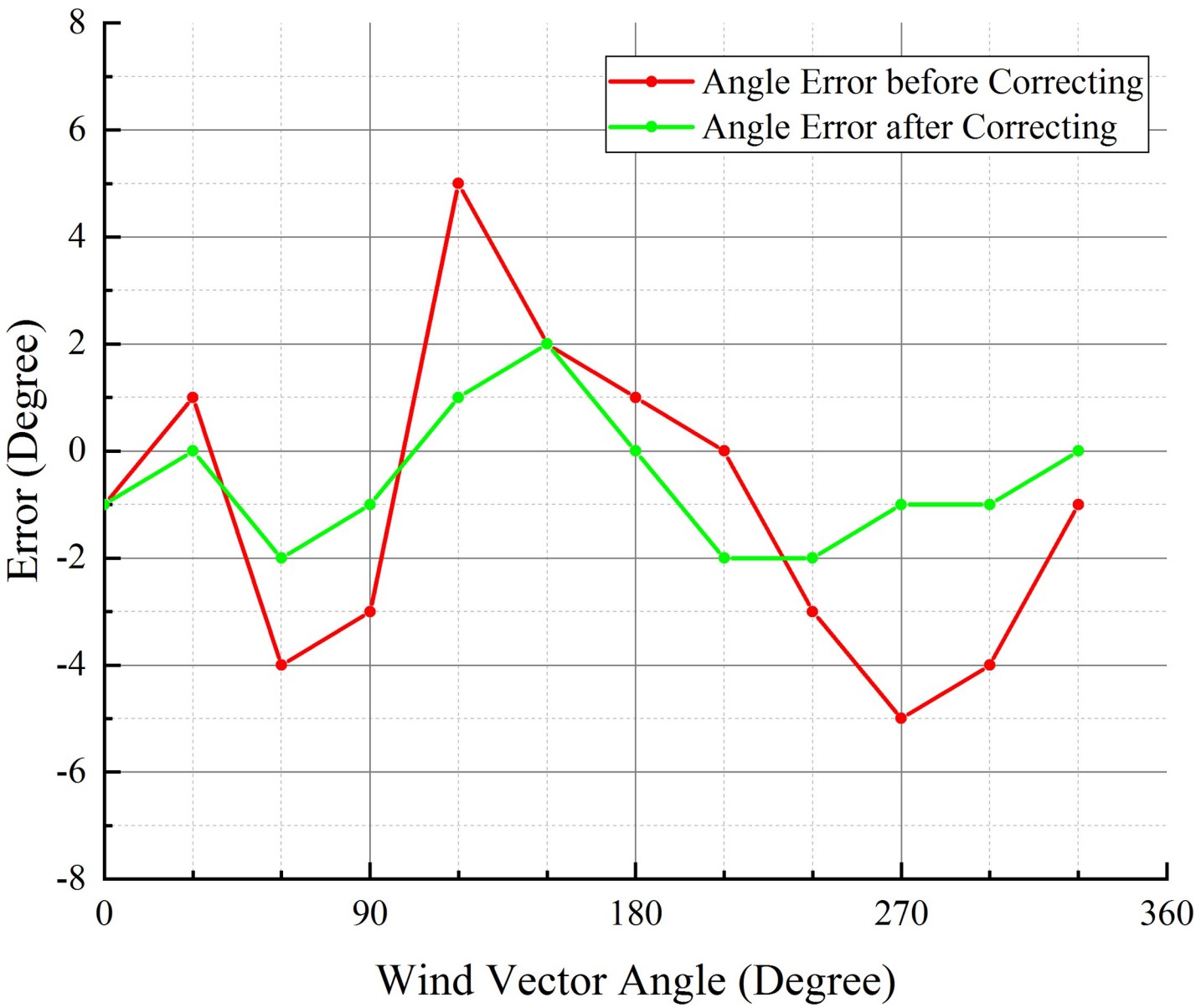

**Fig 17. Comparison of the error distributions of the two wind angle measurement methods.**

**Table 4. Wind angle measurement errors.**

| No. | Preset angle | Measuring angle | Angle error |
|-----|--------------|-----------------|-------------|
| 1 | 8˚ | 8˚ | 0˚ |
| 2 | 21˚ | 21˚ | 0˚ |
| 3 | 53˚ | 53˚ | 0˚ |
| 4 | 96˚ | 95˚ | −1˚ |
| 5 | 124˚ | 123˚ | −1˚ |
| 6 | 172˚ | 172˚ | 0˚ |
| 7 | 193˚ | 195˚ | 2˚ |

(*Continued*)

**Table 4.** (Continued)

| No. | Preset angle | Measuring angle | Angle error |
|-----|-----|-----|-----|
| 8 | 218˚ | 219˚ | 1˚ |
| 9 | 262˚ | 262˚ | 0˚ |
| 10 | 293˚ | 292˚ | −1˚ |

improved. Subsequently, the wind direction measurement accuracy can be improved. Therefore, we will focus on this aspect in future research.

## Supporting information

**S1 Data.**
(ZIP)

**S1 Nomenclature.**
(DOCX)

## Author Contributions

**Conceptualization:** Yunbo Shi.

**Data curation:** Tian Wang.

**Funding acquisition:** Guangdong Lan.

**Investigation:** Tian Wang, Yunbo Shi, Guangdong Lan.

**Methodology:** Tian Wang, Yunbo Shi, Guangdong Lan, Congning Liu.

**Project administration:** Xiaoyu Yu.

**Resources:** Yunbo Shi, Xiaoyu Yu, Congning Liu.

**Supervision:** Yunbo Shi.

**Validation:** Congning Liu.

**Writing – original draft:** Tian Wang.

**Writing – review & editing:** Tian Wang, Yunbo Shi, Xiaoyu Yu, Guangdong Lan, Congning Liu.

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
