## [Decision Letter · Decision Letter 0]

14 Apr 2021

PONE-D-21-04271

Novel strategy for wide-range wind vector measurement using the hybrid CP/CTD heating mode and sequential measuring and correcting

PLOS ONE

Dear Dr. Yu,

Thank you for submitting your manuscript to PLOS ONE. After careful consideration, we feel that it has merit but does not fully meet PLOS ONE’s publication criteria as it currently stands. Therefore, we invite you to submit a revised version of the manuscript that addresses the points raised during the review process.

We look forward to receiving your revised manuscript.

Kind regards,

Mehrdad Ahmadi Kamarposhti

Academic Editor

PLOS ONE

Journal Requirements:

Additional Editor Comments (if provided):

Reviewers' comments:

Reviewer's Responses to Questions

**Comments to the Author**

1. Is the manuscript technically sound, and do the data support the conclusions?

Reviewer #1: Partly

Reviewer #2: Partly

2. Has the statistical analysis been performed appropriately and rigorously? 

Reviewer #1: N/A

Reviewer #2: N/A

3. Have the authors made all data underlying the findings in their manuscript fully available?

Reviewer #1: Yes

Reviewer #2: No

4. Is the manuscript presented in an intelligible fashion and written in standard English?

Reviewer #1: No

Reviewer #2: No

5. Review Comments to the Author

Reviewer #1: The first and general comment relates to the written form, as the paper requires a serious improvement in English grammar and spelling. The English should be improved. A deep proofreading is needed, many errors can be seen through the document.

The introduction and the related review of the literature is poor provided. And the structure must be improved for a better understanding of the current state of the art. Also, the drawbacks of the existing methods must be highlighted clearly for justifying the upgrade proposed by the current work. The way the latter work is improving the state of the art must be clarified.

Novelty of the paper is not mentioned obviously. There're a grammatical and syntax errors.

Authors are encouraged to introduce a nomenclature section at the beginning of the manuscript, including all variables, acronyms, indexes and constants defined in the manuscript, in order to make the text more clear and readable.

In the introduction section, the authors need more to introduce the previous literature. A comprehensive paper needs more than 40 references at least.

Figures should be given with better accuracy and described in the paper. Figures must be replaced with high resolution ones.

Authors must be talk about the future work and potential limitations briefly in the Conclusions and Recommendations section.

Variables in the text must be italic.

All those comments are unfortunate when we see the quality of the numerical results. Via those results, the proposed method demonstrates its effectiveness without any doubt. However, the materials for introducing the state of the art and the methodology is deficient and poor.

6. PLOS authors have the option to publish the peer review history of their article (what does this mean?). If published, this will include your full peer review and any attached files.

Reviewer #1: No

Reviewer #2: No

---

## [Author Response · Author response to Decision Letter 0]

8 Jun 2021

Thank you for your kind comments on this paper! 

Regarding the English writing issues, we have consulted a 3rd-party proofreading service provider for full-text checking on grammar and spelling. Many issues have been found and corrected now. 

Regarding the nomenclature section, we have added in the manuscript, and explained all variables in every formula.

Regarding the references, we added more supplemental references and re-arranged in the text. Now a total of 42 papers are referenced.

Regarding the figures, we tweaked the description texts in the manuscript and the resolution have also been tweaked for better showing.

Regarding the future work and potential limitations, we have talked in the last paragraph of the Conclusion part.

---

## [Decision Letter · Decision Letter 1]

24 Jun 2021

Novel strategy for wide-range wind vector measurement using the hybrid CP/CTD heating mode and sequential measuring and correcting

PONE-D-21-04271R1

Dear Dr. Yu,

We’re pleased to inform you that your manuscript has been judged scientifically suitable for publication and will be formally accepted for publication once it meets all outstanding technical requirements.

Kind regards,

Mehrdad Ahmadi Kamarposhti

Academic Editor

PLOS ONE

Additional Editor Comments (optional):

Reviewers' comments:

Reviewer's Responses to Questions

**Comments to the Author**

1. If the authors have adequately addressed your comments raised in a previous round of review and you feel that this manuscript is now acceptable for publication, you may indicate that here to bypass the “Comments to the Author” section, enter your conflict of interest statement in the “Confidential to Editor” section, and submit your "Accept" recommendation.

Reviewer #1: All comments have been addressed

Reviewer #2: All comments have been addressed

2. Is the manuscript technically sound, and do the data support the conclusions?

Reviewer #1: Yes

Reviewer #2: Partly

3. Has the statistical analysis been performed appropriately and rigorously? 

Reviewer #1: Yes

Reviewer #2: Yes

4. Have the authors made all data underlying the findings in their manuscript fully available?

Reviewer #1: (No Response)

Reviewer #2: No

5. Is the manuscript presented in an intelligible fashion and written in standard English?

Reviewer #1: (No Response)

Reviewer #2: Yes

6. Review Comments to the Author

Reviewer #1: All the comments have been addressed in the newer version of the manuscript. I have no further comments

Reviewer #2: I have no further comments. The paper is well written and organized and can be considered for publication in journal.

7. PLOS authors have the option to publish the peer review history of their article (what does this mean?). If published, this will include your full peer review and any attached files.

Reviewer #1: No

Reviewer #2: No

---

## [Editor Report · Acceptance letter]

28 Jun 2021

PONE-D-21-04271R1 

Novel strategy for wide-range wind vector measurement using the hybrid CP/CTD heating mode and sequential measuring and correcting 

Dear Dr. Yu:

I'm pleased to inform you that your manuscript has been deemed suitable for publication in PLOS ONE. Congratulations! Your manuscript is now with our production department. 

Kind regards, 

on behalf of

Dr. Mehrdad Ahmadi Kamarposhti 

Academic Editor

PLOS ONE